# Simplifying DINO via Coding Rate Regularization

Ziyang Wu [1]  Jingyuan Zhang [2]  Druv Pai [1]  Xudong Wang [1]
Chandan Singh [3]  Jianwei Yang [3]  Jianfeng Gao [3]  Yi Ma [1 2 4]

## Abstract

DINO and DINOv2 are two model families being widely used to learn representations from unlabeled imagery data at large scales. Their learned representations often enable state-of-the-art performance for downstream tasks, such as image classification and segmentation. However, they employ many empirically motivated design choices and their training pipelines are highly complex and unstable — many hyperparameters need to be carefully tuned to ensure that the representations do not collapse — which poses considerable difficulty to improving them or adapting them to new domains. In this work, we posit that we can remove most such-motivated idiosyncrasies in the pre-training pipelines, and only need to add an explicit coding rate term in the loss function to avoid collapse of the representations. As a result, we obtain highly simplified variants of the DINO and DINOv2 which we call SimDINO and SimDINOv2, respectively. Remarkably, these simplified models are more robust to different design choices, such as network architecture and hyperparameters, and they learn even higher-quality representations, measured by performance on downstream tasks, offering a Pareto improvement over the corresponding DINO and DINOv2 models. This work highlights the potential of using simplifying design principles to improve the empirical practice of deep learning. Code and model checkpoints are available at `https://github.com/RobinWu218/SimDINO`.

## 1. Introduction

Self-supervised learning (SSL) is the toolkit of choice to learn representations for large datasets of unlabeled images

[1]UC Berkeley [2]TranscEngram [3]Microsoft Research [4]HKU. Correspondence to: Ziyang Wu <zywu@berkeley.edu>.

*Proceedings of the 42nd International Conference on Machine Learning*, Vancouver, Canada. PMLR 267, 2025. Copyright 2025 by the author(s).

(Hadsell et al., 2006; Oord et al., 2018; Wu et al., 2018; Grill et al., 2020; He et al., 2020; Bardes et al., 2021; Chen & He, 2021; Caron et al., 2021; Zhou et al., 2021; He et al., 2022; Assran et al., 2023; Oquab et al., 2023), captioned images (Radford et al., 2021), videos (Feichtenhofer et al., 2022), and text (Radford et al., 2018; Devlin, 2018; Radford et al., 2019; Brown et al., 2020), among other modalities. In the context of image SSL, there are two main approaches: *reconstructive* (He et al., 2022), where the goal is to reconstruct some function of the true image data from a "view", i.e., corruption or augmentation, and *contrastive* (Hadsell et al., 2006), where the goal is, for each image, to have the features of different views of the image all be close, and features of views of different images be far.

Within contrastive SSL, a key challenge lies in preventing *representation collapse*, where models learn trivial solutions that map all inputs to the same output. One common approach to address this is through the use of *negative samples*, which explicitly encourages representations of different images to be dissimilar. Thus far, the success of using negative samples depends on having a large batch size (Wu et al., 2018; He et al., 2020), which poses computational challenges at scale. Methods which attempt to avoid this bottleneck by using negative samples in more implicit and indirect ways to avoid collapse (Caron et al., 2021) can cope with smaller batch sizes, but often require training pipelines with many components and hyperparameters carefully tuned to avoid collapse, making them difficult to train.

The state-of-the-art for image SSL is generally considered to be the DINOv2 model family (Oquab et al., 2023). It is built on the DINO model family (Caron et al., 2021). Both classes of models are trained using contrastive SSL and thus run into the representation collapse issue. While DINOv2 explicitly and directly uses negative samples to avoid collapse, it inherits much of its training pipeline from DINO, which uses negative samples more indirectly. As such, *both* model families' training pipelines are highly complex and unstable, requiring many tweaks and careful hyperparameter selection in order for the training to converge for a given architecture. Despite this capriciousness, the trained models' representations are highly useful for downstream tasks, and are widely used (Baharoon et al., 2023; Wei et al., 2024).

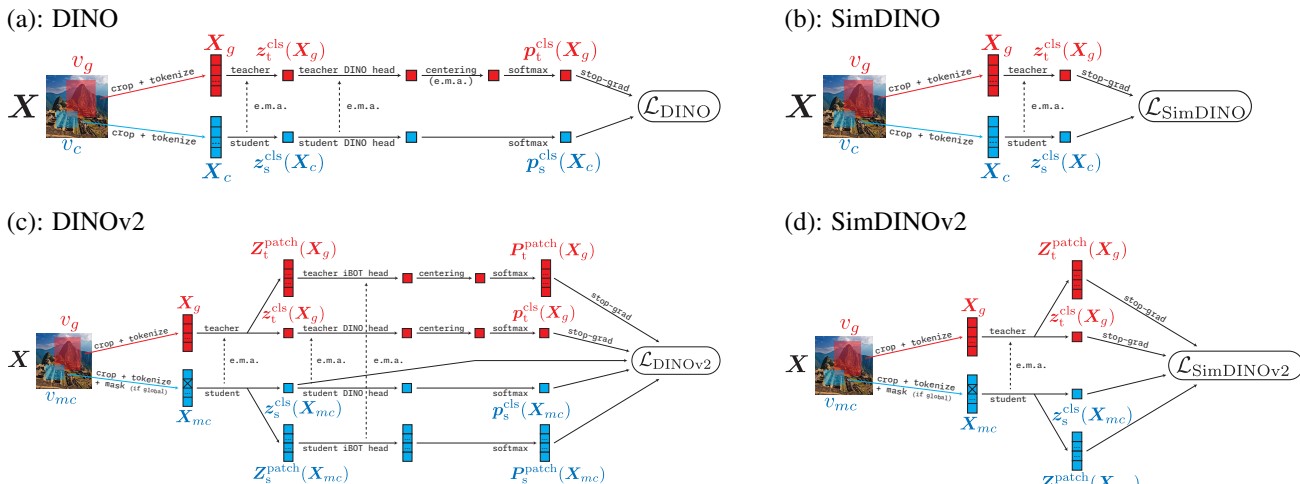

*Figure 1.* **The DINO and DINOv2 pipelines are substantially simplified to the respective SimDINO and SimDINOv2 pipelines.** (a) In the DINO pipeline, an input image is turned into patches. Then a global view $v_g$ and a local view $v_c$ are randomly sampled. The global view is pushed through the teacher encoder, while the other view is through the student encoder. (b) The SimDINO pipeline removes the need for expensive post-processing operations present in DINO, such as a dimension-increasing linear layer and a high-dimensional softmax. (c) The DINOv2 pipeline adds masking (here masked patches are denoted by ×) and an additional loss on image patch features to the DINO pipeline. (d) The SimDINOv2 training operates directly on the learned representations, simplifying the pipeline.

**Our contributions.** In this work, we remove many tweaks and hyperparameters from the DINO and DINOv2 training pipelines, replacing them with a term in the objective which explicitly uses negative samples. We show empirically that this term, which involves the *total coding rate* regularizer (Ma et al., 2007; Yu et al., 2020; Li et al., 2022), enables much more simple, robust, and computationally efficient training pipelines, as shown in Figure 1. We show that the resulting models, named SimDINO and SimDINOv2, learn representations that achieve even higher state-of-the-art performance as those learned by DINO and DINOv2 across a variety of downstream tasks. Our work underscores the value of understanding and simplifying pipelines to improve performance in vision SSL.

**Notation.** Let $C, H, W, D, N, d \geq 1$ be positive integers. For a set $A$, let the space of finite sequences of elements of $A$ be denoted as $A^* = \bigcup_{t=1}^{\infty} A^t$. Our data will be images $\boldsymbol{X} \in \mathbb{R}^{C \times H \times W}$. We consider different augmentations, or *views*, of the input data $\boldsymbol{X}$, such as rotations or crops; we can represent a view as a function $v \colon \mathbb{R}^{C \times H \times W} \to \mathbb{R}^{D \times N_v}$ where $N_v$ is the number of tokens in the view. By an abuse of vocabulary we also call $v(\boldsymbol{X}) \in \mathbb{R}^{D \times N_v}$ a view.

Let $\mathbb{S}_{d-1} \subseteq \mathbb{R}^d$ be the $(d-1)$-dimensional $\ell^2$-sphere. For the purpose of representation learning, we will consider an *encoder* neural network parameterized by weights $\theta \in \Theta$ (i.e., the weight space), as a function $f_\theta \colon (\mathbb{R}^D)^* \to \mathbb{S}_{d-1} \times \mathbb{S}_{d-1}^*$. We factor $f_\theta = (f_\theta^{\mathrm{cls}}, f_\theta^{\mathrm{patch}})$ where $f_\theta^{\mathrm{cls}} \colon \mathbb{R}^{D \times *} \to \mathbb{S}_{d-1}$ outputs the so-called *class token feature* (i.e., an aggregate representation of the input patches) and $f_\theta^{\mathrm{patch}} \colon (\mathbb{R}^D)^* \to \mathbb{S}_{d-1}^*$ outputs the *patch tokens' fea-*

*tures* (i.e., a representation for each input patch). The network is a Vision Transformer (Dosovitskiy, 2020; Touvron et al., 2021) with appended multi-layer perceptrons (MLPs) to post-process each feature followed by $\ell^2$-normalizations.

## 2. Methods: Simplifying DINO and DINOv2

### 2.1. Recap of the Original DINO Pipeline

The goal of DINO is to learn an aggregate representation of the input image which contains information about large-scale semantics of the input (e.g., the locations and properties of different objects in the image). They do this via a pre-training pipeline (Caron et al., 2021) which is depicted in Figure 1(a), and we also describe it throughout this section. The main idea is to take multiple *views* (i.e., different crops) of the data, and ensure that the features generated by these views are consistent with each other (in a sense which will be made precise shortly) as much as possible. If the views each contain a salient part of the input such as a central object, the feature corresponding to any view would then contain information about this central object. The end goal is that the feature of any large-enough view contains information about all relevant objects in the input image, which can then be extracted for use in downstream tasks such as image classification or image segmentation.

In the rest of the section, we will discuss the pre-training pipeline. As is common in contrastive SSL, the DINO framework uses two networks: a so-called *teacher* network parameterized by $\theta_t \in \Theta$, and a so-called *student* network parameterized by $\theta_s \in \Theta$. During pre-training, the loss

will encourage the student's representation to align with the teacher's representation, even as the teacher is simultaneously updated using student weights; this is *self-distillation*, and can be viewed as an optimization strategy or even implicitly regularizing the objective (Chen & He, 2021).

In the pipeline, we process each image $X$ in the following way. First, we sample at random a view, or crop, $v_c$, independently of $X$; the view can *either* be a "global view" (i.e., a large crop) or a "local view" (i.e., small crop). We denote $X_c := v_c(X) \in (\mathbb{R}^{D \times N_{\mathrm{loc}}} \cup \mathbb{R}^{D \times N_{\mathrm{glo}}})$. We also sample a global view $v_g$, and denote $X_g := v_g(X) \in \mathbb{R}^{D \times N_{\mathrm{glo}}}$.[1] Views are implemented in the same way as in DINO; they are formally described in Appendix A for completeness.

The (local or global) view $X_c$ is fed to the student network[2] $f_{\theta_s}^{\mathrm{cls}}$ to get $z_s^{\mathrm{cls}}(X_c) \in \mathbb{R}^d$, while the global view $X_g$ is fed to the teacher network $f_{\theta_t}^{\mathrm{cls}}$ to get $z_t^{\mathrm{cls}}(X_g) \in \mathbb{R}^d$, i.e.:

$$z_s^{\mathrm{cls}}(X_c) := f_{\theta_s}^{\mathrm{cls}}(X_c), \qquad z_t^{\mathrm{cls}}(X_g) := f_{\theta_t}^{\mathrm{cls}}(X_g). \quad (1)$$

Now, it is certainly possible to directly compare and evaluate these features. However, DINO adds post-processing steps, arguing that they improve performance and prevent collapse:

- They add weight-normalized linear layers (Salimans & Kingma, 2016) $h_{\eta_s^{\mathrm{DINO}}}, h_{\eta_t^{\mathrm{DINO}}} : \mathbb{R}^d \to \mathbb{R}^m$ where $m \gg d$, called the "DINO heads" and parameterized by $\eta_s^{\mathrm{DINO}}, \eta_t^{\mathrm{DINO}}$, appended to the end of the student and teacher networks respectively.

- They center the teacher-computed features using a learned vector $\mu \in \mathbb{R}^m$.

- They take a temperature-weighted softmax of both features to compute probability vectors in $\mathbb{R}^m$, sometimes called *prototype scores*, which they then can compare using cross-entropy.

Mathematically, the post-processing steps to get probability vectors for each view are as follows:

$$p_s^{\mathrm{cls}}(X_c) := \mathrm{softmax}(h_{\eta_s^{\mathrm{DINO}}}(z_s^{\mathrm{cls}}(X_c))/\tau_s) \qquad (2)$$

$$p_t^{\mathrm{cls}}(X_g) := \mathrm{softmax}([h_{\eta_t^{\mathrm{DINO}}}(z_t^{\mathrm{cls}}(X_g)) - \mu]/\tau_t) \quad (3)$$

where $\tau_s, \tau_t > 0$ are temperature parameters for the student and teacher respectively. Finally, the loss (to be minimized) encourages $p_s^{\mathrm{cls}}(X_c)$ and $p_t^{\mathrm{cls}}(X_g)$ to be close using a cross-entropy functional $d_{\mathrm{CE}}$, which effectively distills the teacher into the student by aligning the predicted outputs:

$$\mathcal{L}_{\mathrm{DINO}} := \mathbb{E}[d_{\mathrm{CE}}(p_t^{\mathrm{cls}}(X_g), p_s^{\mathrm{cls}}(X_c))] \qquad (4)$$

where the expectation is over $X$, the (local or global) view $v_c$, and the global view $v_g$, and the function $d_{\mathrm{CE}}$ is defined via the cross-entropy as

$$d_{\mathrm{CE}}(p, q) := -\sum_{i=1}^{m} p_i \log q_i. \qquad (5)$$

When training, DINO estimates the expectation in (4) by a stratified plug-in estimator over a batch of sample images. That is, to estimate the expectation, we condition on $X$ then estimate the conditional expectation $\mathbb{E}[d_{\mathrm{CE}}(\cdot, \cdot) \mid X]$ via plug-in using several different global views (usually two global views, which play the role of the arbitrary view $v_c$ and the global view $v_g$) and several different local views, and finally average over $X$ to obtain the estimate. The optimization of this estimated loss, too, is done in an ad-hoc way; while all five parameters $\theta_s, \theta_t, \eta_s^{\mathrm{DINO}}, \eta_t^{\mathrm{DINO}}, \mu$ are updated at each iteration, they update in different ways:

- The student parameters $\theta_s$ and $\eta_s^{\mathrm{DINO}}$ are updated via an iteration of a stochastic gradient descent (SGD)-type algorithm, such as Adam, on the loss (4). The backpropagation for the loss gradient is computed assuming the teacher parameters $\theta_t, \eta_t^{\mathrm{DINO}}$, and $\mu$ are "frozen" or constants (i.e., "stop-gradient").

- The teacher parameters $\theta_t, \eta_t^{\mathrm{DINO}}$, and $\mu$ are updated via exponentially moving averages (EMAs) of the student weights $\theta_s$, the student DINO head $\eta_s^{\mathrm{DINO}}$, and the average output of teacher the DINO head $\mathbb{E}[h_{\eta_t^{\mathrm{DINO}}}(z_t^{\mathrm{cls}}(X_g))]$ (in practice estimated over a minibatch), respectively. Formally, for decay parameters $\lambda, \nu \in [0, 1]$, at each iteration we compute $\theta_t \leftarrow \lambda\theta_t + (1-\lambda)\theta_s, \eta_t^{\mathrm{DINO}} \leftarrow \lambda\eta_t^{\mathrm{DINO}} + (1-\lambda)\eta_s^{\mathrm{DINO}}$, and $\mu \leftarrow \nu\mu + (1-\nu)\mathbb{E}[h_{\eta_t^{\mathrm{DINO}}}(z_t^{\mathrm{cls}}(X_g))]$.

The decay parameters $\lambda, \nu$ and the temperature parameter $\tau$ change along the optimization trajectory, and their schedules are design decisions which impact convergence.

As previously mentioned, many of the ad-hoc methods and choices described above are due to a tension: a *trivial* solution to optimizing (4) is to enforce that $f_{\theta_s}$ and $f_{\theta_t}$ *collapse*, i.e., become or approximate the constant function, which map each local and global view to the same feature $z$ or even to the same probability vector $p$. To explain why DINO does not collapse, we wish to highlight the centering operation in (3), which computes batch statistics during its EMA update, hence using negative samples and implicitly pushing different samples' features apart, even though the precise conceptual mechanism by which this occurs is not clear and involves a careful interaction between the centering vector and temperature scaling (Caron et al., 2021). Indeed, Caron et al. (2021) shows that collapsed solutions are common without very carefully tuning the EMA schedule

---

[1] More precisely, let $c$ be a random vector containing the boundaries of the crop, so that $v_c$ crops exactly the region supplied by $c$. Analogous notation can be defined for $g$ and $v_g$.

[2] Note that the parameters $\theta_s$ and $\theta_t$ each contain a positional encoding over all patches; when a view is fed through the network, it receives an interpolated positional encoding of the view's crop.

and temperature schedule, and arguing that the remaining hyperparameters and choices would severely degrade the performance if perturbed. A more in-depth discussion of the tension, and the added complexity required to train a model in spite of it, is in Appendix B. As we will see, if this tension is alleviated in an alternative way, many hyperparameters can be removed and the rest can be changed robustly.

## 2.2. From DINO to SimDINO

To go from DINO to SimDINO, we ask the question:

*Can we just compare $z_s^{cls}(X_c)$ and $z_t^{cls}(X_g)$?*

If we could do this, then we could avoid the large DINO head, the centering operation, the softmaxes, and the cross-entropy based loss. However, the mechanism in DINO for avoiding representation collapse via negative samples would therefore be removed. Thus, we have a second question:

*Can we efficiently use the negative samples' features explicitly to enforce non-collapse?*

For the first question, we argue that the most simple *squared Euclidean distance*, namely

$$d_{\ell^2}(\boldsymbol{x}, \boldsymbol{y}) := \frac{1}{2}\|\boldsymbol{x} - \boldsymbol{y}\|_2^2 \qquad (6)$$

works at least as well as the cross-entropy-based functional (5) applied to an affine transformation of the features, as in (4). For the second question, we argue that we may directly penalize the covariance of the features in order to avoid collapse, as follows. For a hyperparameter $\varepsilon > 0$, the (total) coding rate (Ma et al., 2007; Yu et al., 2020; Li et al., 2022) of a symmetric positive semidefinite matrix $\boldsymbol{\Gamma} \in \mathbb{R}^{d \times d}$ is

$$R_\varepsilon(\boldsymbol{\Gamma}) := \frac{1}{2}\log\det\left(\boldsymbol{I} + \frac{d}{\varepsilon^2}\boldsymbol{\Gamma}\right), \qquad (7)$$

In words, $R_\varepsilon$ is an approximation to the rate distortion with quantization error $\varepsilon$ of a Gaussian random variable with covariance $\boldsymbol{\Gamma}$ (and this approximation is perfect in the limit $\varepsilon \to 0$). More concretely, it is a measure of size of the covariance, even if the underlying variables are non-Gaussian. Thus one way to ensure non-collapse is to add $-R_\varepsilon(\text{Cov}[z_s^{cls}(X_c) \mid v_c \in \mathcal{V}_{glo}])$ as a regularizer (where $v_c \in \mathcal{V}_{glo}$ means that $v_c$ is a global view),[3] yielding the loss

$$\mathcal{L}_{\text{SimDINO}} := \mathbb{E}[d_{\ell^2}(z_t^{cls}(X_g), z_s^{cls}(X_c))] \qquad (8)$$
$$- \gamma R_\varepsilon(\text{Cov}[z_s^{cls}(X_c) \mid v_c \in \mathcal{V}_{glo}]).$$

---

[3]We only use global views' features for the sake of efficiency. If $d_{\ell^2}$ in (8) is small, then the local and global views' student features are close since they are both close to the global views' teacher features, so $\text{Cov}[z_s^{cls}(X_c)] \approx \text{Cov}[z_s^{cls}(X_c) \mid v_c \in \mathcal{V}_{glo}]$.

where $\gamma > 0$ is a hyperparameter. Note that $d_{\ell^2}(z_t^{cls}, z_s^{cls}) = 1 - (z_s^{cls})^\top z_t^{cls}$ since $z_s^{cls}, z_t^{cls} \in \mathbb{S}_{d-1}$.

When training, similar to DINO, we estimate the expectation and covariance in (8) by a type of plug-in estimator. Namely, the expectation is estimated similar to DINO, just using $d_{\ell^2}$ instead of $d_{CE}$. To estimate the coding rate, we sub-sample several $z_s^{cls}(X_c)$ over both $X$ and $v_c$, estimate $\text{Cov}[z_s^{cls}(X_c) \mid v_c \in \mathcal{V}_{glo}]$ on that sub-sample via plug-in, estimate $R_\varepsilon$ of the population covariance by calculating it on the sample covariance, then average the estimates over all sub-samples. We conjecture that the latter estimator has lower variance compared to the naive plug-in estimator for $\text{Cov}[z_s^{cls}(X_c) \mid v_c \in \mathcal{V}_{glo}]$ as it is similar to variance-reduction methods in statistics (Kahn & Marshall, 1953), which we hypothesize might be a factor as to why SimDINO can handle a smaller batch size than other contrastive SSL methods that explicitly use negative samples but avoid collapse using higher-variance or more implicit regularizers.

The overall pipeline is shown in Figure 1(b). Note that it is much simpler than DINO. We provide pseudocode for the training pipeline in Algorithm 1 in Appendix D.

After training, we use the teacher network for evaluation.

## 2.3. From DINOv2 to SimDINOv2

The pipeline of the DINOv2 framework (Oquab et al., 2023), as shown in Figure 1(c), is built upon the DINO pipeline, and has two main goals: first, learn an *aggregate* representation which contains large-scale semantics of the input (i.e., the goal of DINO); second, learn *patch-based* representations which have fine-grained semantic information about each patch and its local neighborhood. The main new ideas to achieve this, drawn from the iBOT pipeline (Zhou et al., 2021), are that *the input to the student has some masked patches*, and that *the loss also computes similarity of the patch-based features*. To see why this works, consider if some patches are masked, and the model is able to predict masked patches using their unmasked neighbors, then from each patch the model can extract strong information about the semantics of nearby patches, which is an idea similar in spirit to masked autoencoding (He et al., 2022). Thus, these two ideas from iBOT would furnish our model with informative patch-based representations.

We now discuss the DINOv2 pipeline, before discussing our modifications. While we have the same ("base") views $v_c$ and $v_g$ as before, we also consider a *masked view* $v_{mc}$, which computes $v_c$ but, if $v_c \in \mathcal{V}_{glo}$, *subsequently replaces a fraction* $\alpha \in [0, 1]$ *of the tokens in the view output with a learnable mask token* $\boldsymbol{x}_{mask}$ (as in (He et al., 2022), the mask token is shared across all views). Similarly to previous notation, we denote $X_{mc} := v_{mc}(X) \in (\mathbb{R}^{D \times N_{loc}} \cup \mathbb{R}^{D \times N_{glo}})$.

Now that we have this setup, we do similar operations to

DINO pipeline, with some changes:

- There are additional "iBOT heads" for the student and teacher, processing the patch-based features column-wise (i.e., patch-wise), with weights $\eta_{\mathrm{s}}^{\mathrm{iBOT}}, \eta_{\mathrm{t}}^{\mathrm{iBOT}}$ (cf the "DINO head" with weights $\eta_{\mathrm{s}}^{\mathrm{DINO}}, \eta_{\mathrm{t}}^{\mathrm{DINO}}$).

- The centering operation on teacher-output features is performed on both the aggregate features and (column-wise) on the patch-wise features.

- The centering operation uses three iterations of the Sinkhorn-Knopp algorithm (Cuturi, 2013; Caron et al., 2020), denoted below by SKC, instead of an EMA, and is parameter-free but more expensive than simple subtraction. Note that the Sinkhorn-Knopp algorithm uses features from all images in each minibatch.

Let $\boldsymbol{Z}^{\mathrm{patch}}$ and $\boldsymbol{P}^{\mathrm{patch}}$ be the patch-wise representations and prototype scores respectively, and let $\boldsymbol{z}^i \in \mathbb{R}^d$ be the $i^{\mathrm{th}}$ column of $\boldsymbol{Z}^{\mathrm{patch}}$ (and similar for $\boldsymbol{p}^i \to \boldsymbol{P}^{\mathrm{patch}}$). Then we have (where $1 \le i \le N_{\mathrm{glo}}$):

$$(\boldsymbol{z}_{\mathrm{s}}^{\mathrm{cls}}(\boldsymbol{X}_{mc}), \boldsymbol{Z}_{\mathrm{s}}^{\mathrm{patch}}(\boldsymbol{X}_{mc})) := f_{\theta_{\mathrm{s}}}(\boldsymbol{X}_{mc}) \tag{9}$$

$$(\boldsymbol{z}_{\mathrm{t}}^{\mathrm{cls}}(\boldsymbol{X}_g), \boldsymbol{Z}_{\mathrm{t}}^{\mathrm{patch}}(\boldsymbol{X}_g)) := f_{\theta_{\mathrm{t}}}(\boldsymbol{X}_g) \tag{10}$$

$$\boldsymbol{p}_{\mathrm{s}}^{\mathrm{cls}}(\boldsymbol{X}_{mc}) := \mathrm{softmax}(h_{\eta_{\mathrm{s}}^{\mathrm{DINO}}}(\boldsymbol{z}_{\mathrm{s}}^{\mathrm{cls}}(\boldsymbol{X}_{mc}))/\tau_{\mathrm{s}}) \tag{11}$$

$$\boldsymbol{p}_{\mathrm{s}}^{i}(\boldsymbol{X}_{mc}) := \mathrm{softmax}(h_{\eta_{\mathrm{s}}^{\mathrm{iBOT}}}(\boldsymbol{z}_{\mathrm{s}}^{i}(\boldsymbol{X}_{mc}))/\tau_{\mathrm{s}}) \tag{12}$$

$$\boldsymbol{p}_{\mathrm{t}}^{\mathrm{cls}}(\boldsymbol{X}_g) := \mathrm{softmax}(\mathrm{SKC}[h_{\eta_{\mathrm{t}}^{\mathrm{DINO}}}(\boldsymbol{z}_{\mathrm{t}}^{\mathrm{cls}}(\boldsymbol{X}_g))]/\tau_{\mathrm{t}}) \tag{13}$$

$$\boldsymbol{p}_{\mathrm{t}}^{i}(\boldsymbol{X}_g) := \mathrm{softmax}(\mathrm{SKC}[h_{\eta_{\mathrm{t}}^{\mathrm{iBOT}}}(\boldsymbol{z}_{\mathrm{t}}^{i}(\boldsymbol{X}_g))]/\tau_{\mathrm{t}}). \tag{14}$$

We then compute the loss using all such probability vectors:

$$\mathcal{L}_{\mathrm{DINOv2}} := \frac{1}{2} \mathbb{E}[d_{\mathrm{CE}}(\boldsymbol{p}_{\mathrm{t}}^{\mathrm{cls}}(\boldsymbol{X}_g), \boldsymbol{p}_{\mathrm{s}}^{\mathrm{cls}}(\boldsymbol{X}_{mc}))] \tag{15}$$

$$+ \frac{1}{2} \mathbb{E}\left[ \frac{1}{N_{\mathrm{glo}}} \sum_{i=1}^{N_{\mathrm{glo}}} d_{\mathrm{CE}}(\boldsymbol{p}_{\mathrm{t}}^{i}(\boldsymbol{X}_g), \boldsymbol{p}_{\mathrm{s}}^{i}(\boldsymbol{X}_{mc}))\mathbf{1}_{i,mc} \,\middle|\, v_c = v_g \right]$$

$$- \gamma \, \mathrm{Entropy}(\boldsymbol{z}_{\mathrm{s}}^{\mathrm{cls}}(\boldsymbol{X}_{mc}) \mid v_c \in \mathcal{V}_{\mathrm{glo}}),$$

where $\mathbf{1}_{i,mc}$ is 1 if patch $i$ of $\boldsymbol{X}_c$ is masked by $v_{mc}$ and 0 otherwise, and the Entropy functional is the differential entropy; it plays a similar role as the coding rate $R_\varepsilon$ in SimDINO (and shortly SimDINOv2) in ensuring non-collapse. It is estimated by Oquab et al. (2023) using the KoLeo estimator (Delattre & Fournier, 2017)) which explicitly uses negative samples. However, the KoLeo estimator is a non-parametric estimator of the expectation of a function of a high-dimensional probability density (Beirlant et al., 1997), and so it has relatively poor sample efficiency (i.e., the required batch size to converge in practice is large).

**We now greatly simplify the above pipeline** using the same ideas as introduced in SimDINO. Namely, we dispense with the DINO/iBOT heads, the Sinkhorn-Knopp centering, and

the softmaxes, and compute the Euclidean distance-based loss directly on normalized features. We obtain the loss

$$\mathcal{L}_{\mathrm{SimDINOv2}} := \frac{1}{2} \mathbb{E}[d_{\ell^2}(\boldsymbol{z}_{\mathrm{t}}^{\mathrm{cls}}(\boldsymbol{X}_g), \boldsymbol{z}_{\mathrm{s}}^{\mathrm{cls}}(\boldsymbol{X}_{mc}))] \tag{16}$$

$$+ \frac{1}{2} \mathbb{E}\left[ \frac{1}{N_{\mathrm{glo}}} \sum_{i=1}^{N_{\mathrm{glo}}} d_{\ell^2}(\boldsymbol{z}_{\mathrm{t}}^{i}(\boldsymbol{X}_g), \boldsymbol{z}_{\mathrm{s}}^{i}(\boldsymbol{X}_{mc}))\mathbf{1}_{i,mc} \,\middle|\, v_c = v_g \right]$$

$$- \gamma R_\varepsilon(\mathrm{Cov}[\boldsymbol{z}_{\mathrm{s}}^{\mathrm{cls}}(\boldsymbol{X}_{mc}) \mid v_c \in \mathcal{V}_{\mathrm{glo}}])$$

The same caveats as in SimDINO apply with respect to how the expectations and covariances are estimated, and the optimization and evaluation procedures carry over. We provide pseudocode for the training pipeline in Algorithm 2 in Appendix D. In the sequel, we will show that these greatly simplified designs actually help the model performance.

**Optimal value for $\gamma$.** In both the SimDINO loss (8) and the SimDINOv2 loss (16), in order to aid learning while making sure neither the distance term nor the regularizer term dominates, we choose $\gamma$ up to an absolute constant factor so that it balances the asymptotic order of the gradient (Frobenius) norms of both terms. By the Cauchy-Schwarz inequality, it suffices to equalize the norms of the gradients of each term w.r.t. the features $\boldsymbol{Z}$. Since the features are normalized on the sphere, the gradient (Frobenius) norm of the distance term is $\mathcal{O}(1)$. For the second term, assuming that we use a batch size $B$, the gradient norm of the second term is $\mathcal{O}(\sqrt{d\min\{d, B\}/B})$. To make these equivalent, we take $\gamma = \Theta(\sqrt{B/(d\min\{d, B\})})$. The same rate holds for SimDINOv2. While this choice of $\gamma$ is ultimately a heuristic, and the constant factor needs to be tuned, it helps to scale SimDINO and SimDINOv2 in practice. Formal calculations, including prescriptive choice for $\gamma$ taking all parameters into account, are provided in Theorems C.1 and C.2.

## 3. Experimental Verification

In this section, we empirically investigate and evaluate our proposed SimDINO and SimDINOv2 models and compare them to the original DINO and DINOv2 model families. In particular, we examine their differences in training dynamics and learned representation both quantitatively and qualitatively. Overall, our experiments show that our proposed SimDINO model families can achieve better performance and learn representations of higher quality than the original DINO families while being significantly simpler and more robust to variations in hyperparameters and architecture.

### 3.1. Experimental Setup

**Model architecture.** Since our method is directly built upon DINO and DINOv2, we adopt settings as close as possible to the original method for fair comparison. Specifically, for all inputs we set patch size to be 16; we use the small, base, and large models of the ViT (Dosovitskiy, 2020) architecture as the backbone, which is connected to a projector

composed of three MLP layers with a hidden size of 2048 and an output dimension of 256. The output features after the projector are $\ell^2$ normalized. Specifically for original (i.e., unsimplified) DINO models, these normalized features are then fed to a weight-normalized linear layer that outputs a high-dimensional (e.g., 65536) vector, before computing the softmax and then the cross-entropy loss.

**Datasets and optimization.** For pretraining, we use the ImageNet-1K dataset across all methods. For fair comparison, we closely follow the original works (Caron et al., 2021; Oquab et al., 2023). We choose AdamW (Loshchilov, 2017) as the optimizer and adopt the same optimization strategies (e.g., learning rates, warm-up schedules). For multicrop augmentation, we use 10 local views of resolution $96 \times 96$ and 2 global views of resolution $224 \times 224$ for all experiments. We provide more details on hyperparameter choices in Appendix E. We also consider several downstream tasks. Specifically, we evaluate our pretrained models on 1) unsupervised object detection and segmentation on COCO val2017 (Lin et al., 2014), 2) semantic segmentation on ADE20K (Zhou et al., 2017), and 3) video object segmentation on DAVIS-2017 (Pont-Tuset et al., 2017).

### 3.2. Experimental Results

**ImageNet Classification.** We report the classification accuracies on ImageNet-1k in Table 1. Following (Caron et al., 2021), we evaluate both $k$-NN and linear accuracy on the ViT backbones pretrained by the DINO model families and our simplified variants. We observe that under both DINO and DINOv2 paradigms, our simplified methods are able to outperform the original pipelines. Furthermore, we observe that applying identical hyperparameter settings from ViT-B to ViT-L results in instability and divergence in DINO, while the same setup yields a steady improvement for SimDINO. To better understand the optimization dynamics of SimDINO, we visualize the evolution of accuracy during training in Figure 2. It can be observed that performance of SimDINO steadily improves as training progresses, while optimization of DINO noticeably slows down, with even a slight performance drop near the end of training. Together, these results demonstrate our simplified pipelines' stability and ease of optimization compared to the originals.

**Object Detection and Segmentation.** To better understand the learned representation, we evaluate the pretrained models on segmentation and object detection tasks. Specifically, we adopt MaskCut (Wang et al., 2023), an effective unsupervised approach of extracting features from a frozen vision backbone for object detection and instance segmentation. In Figure 3, we present qualitative segmentation results by applying MaskCut on models trained with both DINO and SimDINO. Both methods are observed to produce meaningful segmentation results, confirming the

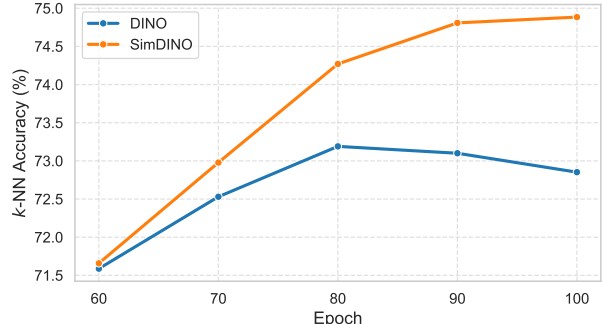

*Figure 2.* **Evolution of $k$-NN accuracy** of ViT-B trained for 100 epochs using DINO and SimDINO paradigms on ImageNet-1K. We omit earlier epochs of similar metrics for better visual clarity.

*Table 1.* **Performance comparison on ImageNet-1K.** SimDINO and SimDINOv2 consistently outperform the original DINO and DINOv2 model families. They are also more stable, while training of DINO on ViT-L diverged (row 3).

| Method | Model | Epochs | $k$-NN | Linear |
|---|---|---|---|---|
| DINO | ViT-B | 100 | 72.9 | 76.3 |
| SimDINO | ViT-B | 100 | **74.9** | **77.3** |
| DINO | ViT-L | 100 | – | – |
| SimDINO | ViT-L | 100 | **75.6** | **77.4** |
| DINOv2 | ViT-B | 100 | 76.0 | 77.2 |
| SimDINOv2 | ViT-B | 100 | **78.1** | **79.7** |
| DINOv2 | ViT-L | 100 | 80.8 | 82.0 |
| SimDINOv2 | ViT-L | 100 | **81.1** | **82.4** |
| SwAV | ViT-S | 800 | 66.3 | 73.5 |
| MoCov3 | ViT-B | 300 | – | 76.7 |

emerging properties similar to the original DINO when using our simplified algorithm. More qualitative results are available in Appendix F.6. To quantitatively evaluate these representation, we perform MaskCut on the COCO val2017 dataset and report our results in Table 2. These results show SimDINO achieves much stronger performance on segmentation and detection tasks than DINO when trained on the same network (row 2 vs 3), and overall even outperforms DINO trained on a smaller patch size[4] (row 2 vs 4).

**Semantic Segmentation on ADE20K.** We evaluate our proposed methods on the ADE20K semantic segmentation task and report the results in Table 3 (column 3 & 4). Specifically, we follow the linear evaluation protocol of (Zhou et al., 2021), where we fix the pretrained backbone and only finetune a linear layer on top of it. From the results, we observe that our proposed SimDINO consistently outperforms the original algorithms. In particular, on ViT-B, SimDINOv2 is able to improve DINOv2 by $4.4@$mIoU.

---

[4]When trained using DINO, ViT models with smaller patch sizes tend to outperform those with larger ones on various tasks including segmentation (Wang et al., 2023; Caron et al., 2021).

*Table 2.* **Unsupervised object detection and segmentation via MaskCut evaluated on COCO val2017** under COCO's official evaluation protocol. SimDINO conclusively performs better than the DINO at detection and segmentation metrics, comparable with DINO with smaller path size (16 vs 8).

| Method | Model | Detection ↑ | | | Segmentation ↑ | | |
|---|---|---|---|---|---|---|---|
| | | $AP_{50}$ | $AP_{75}$ | AP | $AP_{50}$ | $AP_{75}$ | AP |
| SimDINO | ViT-L/16 | **5.4** | 1.9 | 2.4 | 4.5 | 1.4 | 1.9 |
| SimDINO | ViT-B/16 | 5.2 | **2.0** | **2.5** | **4.7** | **1.5** | **2.0** |
| DINO | ViT-B/16 | 3.9 | 1.5 | 1.8 | 3.1 | 1.0 | 1.4 |
| DINO | ViT-B/8 | 5.1 | 2.3 | 2.5 | 4.1 | 1.3 | 1.8 |

These results suggest that our simplified methods lead to representations favorable to dense prediction tasks.

**DAVIS Video Object Segmentation.** In Table 3, we also provide evaluation results on DAVIS-2017 video instance segmentation benchmark. We follow the same evaluation protocol as in (Caron et al., 2021) and segment scenes between consecutive video frames with nearest neighbor. We observe that our proposed SimDINO(v2) outperforms the original methods on this task. One interesting observation is that despite achieving much better $k-$NN accuracy, DINOv2 generally underperforms the original DINO in this task (and similarly for the simplified variants). A similar phenomenon is noted in (Zhou et al., 2021), where this discrepancy is found to be caused by the sensitivity of the evaluation protocol itself (e.g., image resolution). In our evaluation, we do not tune these individual factors and simply adopt the same setting across all models we consider.

**More on Stability and Robustness.** Apart from the observed divergence on ViT-L in Table 1, we note that DINO is sensitive to its pipeline-specific hyperparameters, as evidenced in Table 6 (in Appendix F). To further verify the stability of SimDINO, we experiment with training both algorithms on a different dataset than ImageNet-1k. Specifically, we train them on COCO train2017 (roughly 1/10-th the size of ImageNet-1k), and report the results in Figure 4. Under this setting, SimDINO vastly outperforms DINO. We provide additional ablations on other factors (e.g. batch sizes) in Appendix F. Together, these results demonstrate the superior stability and robustness of SimDINO.

## 4. Related Work

In this section, we identify several previous works which the SimDINO and SimDINOv2 methodologies are similar to or build on. We have already discussed similarities to DINO and DINOv2 in depth so we omit this comparison.

**Siamese contrastive SSL.** Siamese contrastive learning, archetyped by SimCLR (Chen et al., 2020) and SimSiam (Chen & He, 2021) among others, uses the same network

*Table 3.* **Semantic segmentation on ADE20K and video object segmentation on DAVIS-2017.** For semantic segmentation, we train a linear layer on the frozen pretrained backbone. On DAVIS, we segment scenes between video frames using nearest neighbor search. On both tasks, SimDINO(v2) consistently outperforms their original counterparts.

| Method | Model | Lin. Seg. ↑ | | Vid. Seg. ↑ | | |
|---|---|---|---|---|---|---|
| | | mIoU | mAcc | $(\mathcal{J}\&\mathcal{F})_m$ | $\mathcal{J}_m$ | $\mathcal{F}_m$ |
| DINO | ViT-B/16 | 33.1 | 41.9 | 63.0 | 61.5 | 64.4 |
| SimDINO | ViT-B/16 | **33.7** | **42.8** | **63.0** | **61.6** | **64.4** |
| DINOv2 | ViT-B/16 | 32.5 | 41.4 | 53.2 | 52.7 | 53.7 |
| SimDINOv2 | ViT-B/16 | **36.9** | **46.5** | **60.9** | **60.4** | **61.4** |
| DINOv2 | ViT-L/16 | 41.0 | 50.8 | 62.0 | 61.7 | 62.3 |
| SimDINOv2 | ViT-L/16 | **41.8** | **52.2** | **62.6** | **61.9** | **63.3** |

to encode different augmentations (i.e., views) of the same input, and pushes the features of these augmentations together, similar to SimDINO. SimCLR uses explicit negative samples in the loss, while SimSiam manipulates the loss gradient structure using stop-gradients to avoid collapse. Both methods' losses measure alignment or difference via the squared Euclidean distance (equivalently the dot product) of the features. In contrast, SimDINO uses two separate networks — the teacher and student — that update via self-distillation. Furthermore, SimDINO uses the inner product of features in the loss, but it also uses a coding rate regularizer instead of implicitly contrasting negative samples or using the more bespoke contrastive loss in SimCLR.

**Explicit covariance regularization in SSL.** There have also been works that use explicit penalization of the first- and second-order statistics of the features, such as VICReg (Bardes et al., 2021). VICReg uses completely separate networks to encode two augmentations of the same input batch, and then explicitly penalizes the alignment of those features (via Euclidean distance) as well as the features' variance and covariance within the batch, aiming to whiten the features as much as possible. In spirit, this is similar to SimDINO, which also penalizes the alignment and the features' covariance, albeit using a different covariance regularizer and not penalizing the features' variance. Also, SimDINO uses self-distillation to train the teacher network, while VICReg uses two separate networks.

**Self-distillation in SSL.** Several works such as MoCo (He et al., 2020) and BYOL (Grill et al., 2020) train two networks, a teacher and a student, via self-distillation by setting the teacher weights to be an exponential moving average of the student weights. While MoCo uses explicit negative samples from previous batches in its InfoNCE loss computed on a given batch, BYOL does not use negative samples but instead manipulates the gradient structure (akin to SimSiam) in order to prevent collapse, and it uses an extra ("prediction") module appended to the student network,

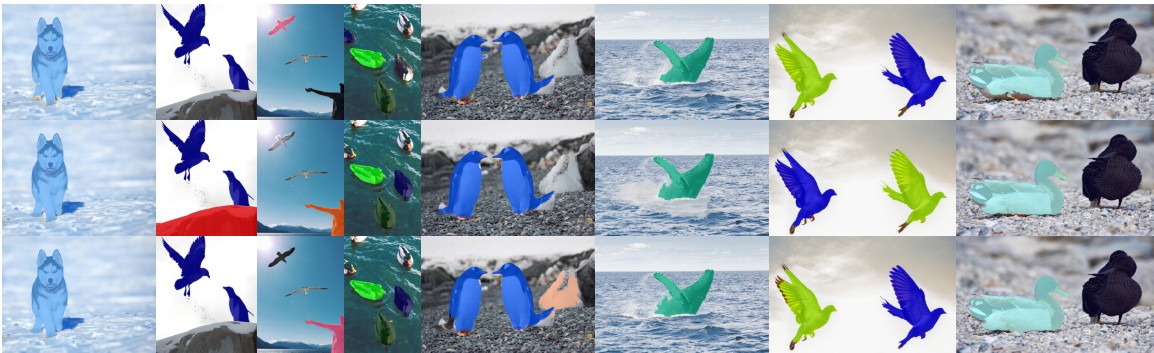

*Figure 3.* **Visualization of MaskCut segmentation results** from DINO ViT-B/16 (row 1), SimDINO ViT-B/16 (row 2) and SimDINO ViT-L/16 (row 3) on selected images.

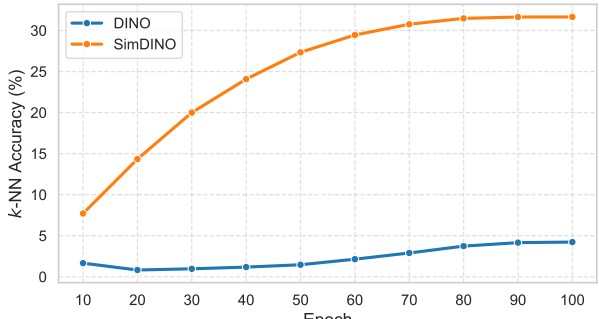

*Figure 4.* $k$-**NN accuracy on ImageNet-1K** of ViT-B trained on **COCO train2017** using DINO and SimDINO paradigms.

making the teacher and student asymmetric. SimDINO uses self-distillation with the same architecture for teacher and student, explicitly uses the simple Euclidean distance in the loss, and explicitly uses the coding rate to prevent collapse.

**Patch feature prediction in SSL.** While most contrastive SSL methods pick a single feature vector (say, of the cls token) as the representation, recent contrastive learning approaches such as DINOv2, I-JEPA (Assran et al., 2023), and C-JEPA (Mo & Tong, 2024) compute losses on the features corresponding to each patch. In I-JEPA, there is one local and one global view, whose crops are nested, and the (Euclidean distance) loss is only computed on the patch features. C-JEPA adds a VICReg-esque variance and covariance penalty to the objective of I-JEPA. In contrast, in SimDINOv2, there are multiple local and global views, the loss incorporates both patch-based and aggregate features, and collapse is prevented by using a coding rate term.

**Coding rate, and related regularizers.** Several works have used coding rate-related terms in the objective (Ma et al., 2007; Yu et al., 2020; Dai et al., 2022; Tong et al., 2022) as well as a way to evaluate quality of representations (Yu et al., 2023; Pai et al., 2023; Wu et al., 2024; Yang et al., 2024). The coding rate has thus been shown to provide a

powerful measure for non-collapse or expansion of the features from a given batch. Other regularizers to accomplish this include the VICReg-type regularizers and the MMCR regularizer (Yerxa et al., 2023; Schaeffer et al., 2024).

## 5. Conclusion

In this work, we identify that the reasons for many empirically motivated design choices in the original DINO and DINOv2 are to avoid collapse of the learned representation. We show that these complicated design choices can be significantly reduced or simplified by adding a coding-rate-related regularization term. The resulting simplified models, called SimDINO and SimDINOv2, are even better in terms of performance for downstream tasks, and their pretraining pipelines are much more robust to different settings and hyperparameters, offering a Pareto improvement against the DINO and DINOv2 model families. Our work demonstrates the value of simplifying deep learning pipelines as well as making tradeoffs as explicit as possible when designing high-performance vision SSL models.

In light of these overarching contributions, there are several possible opportunities for future work. On the theoretical side, our simplified framework provides an entry point for studying the geometric properties of the global optima of self-supervised learning losses. Further study in Appendix F.4 shows that in the framework of the paper, it is possible to set up a self-supervised objective that does not require self-distillation to optimize, making a theoretical analysis much easier, while the resulting model is still quite powerful and practically usable. On the empirical side, one can apply the paradigm of making implicit design choices more explicitly present in the loss to more self-supervised learning frameworks, making existing pipelines more stable and the resulting models of better performance.

## Impact Statement

This paper aims to advance the field of Machine Learning. Since DINO and DINO-2 are very popular methods for unsupervised learning of visual representations, we believe that the methods presented in this work will significantly advance the practice of visual representation learning. Except for values to academics, we do not anticipate any significant social or ethical implications.

## Acknowledgment

Yi Ma would like to acknowledge support from the joint Simons Foundation-NSF DMS grant #2031899, the ONR grant N00014-22-1-2102, the NSF grant #2402951, and also support from and the HKU startup, the Hong Kong Center for Construction Robotics Limited (HKCRC) Award 052245, and JC Club of Hong Kong. Druv Pai acknowledges support from a UC Berkeley College of Engineering Fellowship.

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

## A. Formal Description of Local and Global Views

Each local view, say $v_\ell$ acts as follows, given an input image $\boldsymbol{X}$ of shape $(C, H, W)$. First, for a hyperparameter $p_{\mathrm{loc}} \in [0, 1]$ it crops a rectangular component from $\boldsymbol{X}$ of shape $(C, H_\ell, W_\ell)$, where $H_\ell$ and $W_\ell$ are chosen such that $H_\ell W_\ell = p_{\mathrm{loc}} H W$, i.e., the crop is a fraction $p_{\mathrm{loc}}$ of the whole image. Then the component is resized to shape $(C, S_{\mathrm{loc}}, S_{\mathrm{loc}})$, where $S_{\mathrm{loc}}$ is a hyperparameter, and then divided into $N_{\mathrm{loc}} := S_{\mathrm{loc}}^2 / P^2$ square patches of shape $(C, P, P)$, where the patch size $P$ is a hyperparameter. Each patch is unrolled into a vector of length $D := CP^2$, and the $N_{\mathrm{loc}}$ unrolled vectors are placed in raster order as columns to get the output $\boldsymbol{X}_\ell \in \mathbb{R}^{D \times N_{\mathrm{loc}}}$. Each global view $v_g$ acts the same as a local view, except that the corresponding hyperparameters $p_{\mathrm{glo}}, S_{\mathrm{glo}}$ are larger than their local counterparts $p_{\mathrm{loc}}, S_{\mathrm{loc}}$ (hence also $N_{\mathrm{glo}}$ vs. $N_{\mathrm{loc}}$), while the patch size $P$ (hence dimension $D$) remains the same.[5]

We use these local and global views for training. For evaluation or inference, we do a similar procedure: given $\boldsymbol{X}$ of shape $(C, H, W)$, we resize $\boldsymbol{X}$ proportionally so that its *shorter* edge is length $L_{\mathrm{eval}}$, then take a square crop from the center of shape $(C, S_{\mathrm{eval}}, S_{\mathrm{eval}})$. This sequence is divided into $N_{\mathrm{eval}} := S_{\mathrm{eval}}^2 / P^2$ square patches of length $(C, P, P)$; each patch is unrolled into a vector of length $D := CP^2$, and the $N_{\mathrm{eval}}$ unrolled vectors are placed in raster order as columns to get the output $\boldsymbol{X}_e \in \mathbb{R}^{D \times N_{\mathrm{eval}}}$.

## B. Complex Interactions in DINO and Their Removal

We wish to showcase a finer point about why the DINO pipeline is so unstable. Notice that

$$d_{\mathrm{CE}}(\boldsymbol{p}, \boldsymbol{q}) = -\sum_{i=1}^{m} p_i \log q_i \tag{17}$$

$$= \sum_{i=1}^{m} p_i \log(p_i / q_i) - \sum_{i=1}^{m} p_i \log p_i \tag{18}$$

$$= d_{\mathrm{KL}}(\boldsymbol{p}, \boldsymbol{q}) + H(\boldsymbol{p}) \tag{19}$$

where $d_{\mathrm{KL}}$ is the KL divergence, and $H$ is the entropy of a probability distribution. In other words, this objective is minimized whenever $\boldsymbol{p} = \boldsymbol{q}$ and both are one-hot vectors. Now consider the DINO objective:

$$\mathcal{L}_{\mathrm{DINO}} = \mathbb{E}[d_{\mathrm{CE}}(\boldsymbol{p}_{\mathrm{t}}^{\mathrm{cls}}(\boldsymbol{X}_g), \boldsymbol{p}_{\mathrm{s}}^{\mathrm{cls}}(\boldsymbol{X}_c))] = \mathbb{E}[d_{\mathrm{KL}}(\boldsymbol{p}_{\mathrm{t}}^{\mathrm{cls}}(\boldsymbol{X}_g), \boldsymbol{p}_{\mathrm{s}}^{\mathrm{cls}}(\boldsymbol{X}_c)) + H(\boldsymbol{p}_{\mathrm{t}}^{\mathrm{cls}}(\boldsymbol{X}_g))]. \tag{20}$$

Suppose that, for example, $h_{\eta_{\mathrm{s}}^{\mathrm{DINO}}}$ and $h_{\eta_{\mathrm{t}}^{\mathrm{DINO}}}$ had ranges as a multiple of the all-ones vector, and $\boldsymbol{\mu}$ were a constant multiple of the ones vector. Then the first term in the loss would be minimized, but the second term would become as large as possible (since both $\boldsymbol{p}^{\mathrm{cls}}$ would be just $\frac{1}{m}\boldsymbol{1}_m$, i.e., probability vectors corresponding to the uniform distribution), so this would not be the optimal solution in general. This implies that the learned $h_{\eta_{\mathrm{s}}^{\mathrm{DINO}}}$ and $h_{\eta_{\mathrm{t}}^{\mathrm{DINO}}}$ in general would not both be degenerate. This enables the tradeoff between the EMA parameter $\lambda$ and the temperature parameters $\tau_{\mathrm{s}}, \tau_{\mathrm{t}}$ which enables non-collapse. If the objective just involved the KL divergence and not the entropy term, or else had $h_{\eta_{\mathrm{s}}^{\mathrm{DINO}}}$ be degenerate (manually set and frozen, for instance), or else didn't have a carefully set tradeoff between $\lambda, \tau_{\mathrm{s}}, \tau_{\mathrm{t}}$, then the model would collapse. However, SimDINO removes all of this complexity and replaces it with an explicit coding-rate-type term.

## C. Theory for Hyperparameter Scaling

To develop estimates for how different terms in the loss scale with different batch sizes, we first introduce an empirical version of the SimDINO loss. For a given optimization step, we:

- Sample a minibatch $\{\boldsymbol{X}_1, \ldots, \boldsymbol{X}_B\}$.

- For each sample $\boldsymbol{X}_b$, draw $M_{\mathrm{glo}}$ global views $v_{\mathrm{glo}}^i$ and $M_{\mathrm{loc}}$ local views $v_{\mathrm{loc}}^j$. Define $\boldsymbol{X}_{b,\mathrm{glo}}^i := v_{\mathrm{glo}}^i(\boldsymbol{X}_b)$ and $\boldsymbol{X}_{b,\mathrm{loc}}^j := v_{\mathrm{loc}}^j(\boldsymbol{X}_b)$.

- Compute the features $\boldsymbol{z}_{\mathrm{s}}^{\mathrm{cls}}(\boldsymbol{X}_{b,\mathrm{glo}}^i)$, $\boldsymbol{z}_{\mathrm{s}}^{\mathrm{cls}}(\boldsymbol{X}_{b,\mathrm{loc}}^j)$, and $\boldsymbol{z}_{\mathrm{t}}^{\mathrm{cls}}(\boldsymbol{X}_{b,\mathrm{glo}}^i)$.

---

[5]Of course, we also need the patch size $P$ to divide both the image sizes $S_{\mathrm{loc}}$ and $S_{\mathrm{glo}}$.

- Form the tensors $\boldsymbol{Z}_{\mathrm{s}}^{\mathrm{cls}} \in \mathbb{R}^{B \times M \times d}$, $\boldsymbol{Z}_{\mathrm{t}}^{\mathrm{cls}} \in \mathbb{R}^{B \times M_{\mathrm{glo}} \times d}$ where $M := M_{\mathrm{glo}} + M_{\mathrm{loc}}$, by

$$(\boldsymbol{Z}_{\mathrm{s}}^{\mathrm{cls}})_{bi} = \begin{cases} \boldsymbol{z}_{\mathrm{s}}^{\mathrm{cls}}(\boldsymbol{X}_{b,\mathrm{glo}}^i), & \text{if } 1 \leq i \leq M_{\mathrm{glo}}, \\ \boldsymbol{z}_{\mathrm{s}}^{\mathrm{cls}}(\boldsymbol{X}_{b,\mathrm{loc}}^{i-M_{\mathrm{glo}}}), & \text{otherwise} \end{cases}, \qquad (\boldsymbol{Z}_{\mathrm{t}}^{\mathrm{cls}})_{bi} = \boldsymbol{z}_{\mathrm{t}}^{\mathrm{cls}}(\boldsymbol{X}_{b,\mathrm{glo}}^i) \tag{21}$$

- Compute the loss

$$\widehat{\mathcal{L}}_{\mathrm{SimDINO}} = \frac{1}{2BMM_{\mathrm{glo}}} \sum_{b=1}^{B} \sum_{i=1}^{M_{\mathrm{glo}}} \sum_{j=1}^{M} \|(\boldsymbol{Z}_{\mathrm{t}}^{\mathrm{cls}})_{bi} - (\boldsymbol{Z}_{\mathrm{s}}^{\mathrm{cls}})_{bj}\|_2^2 + \frac{\gamma}{M_{\mathrm{glo}}} \sum_{i=1}^{M_{\mathrm{glo}}} R_\varepsilon\left(\frac{(\boldsymbol{Z}_{\mathrm{s}}^{\mathrm{cls}})_{:,i}^\top (\boldsymbol{Z}_{\mathrm{s}}^{\mathrm{cls}})_{:,i}}{B}\right), \tag{22}$$

where for $1 \leq i \leq M_{\mathrm{glo}}$ we have $(\boldsymbol{Z}_{\mathrm{s}}^{\mathrm{cls}})_{:,i} = [(\boldsymbol{Z}_{\mathrm{s}}^{\mathrm{cls}})_{1,i}, \ldots, (\boldsymbol{Z}_{\mathrm{s}}^{\mathrm{cls}})_{B,i}]^\top \in \mathbb{R}^{B \times d}$.

Our main theorem for SimDINO is the following.

**Theorem C.1.** *The (Frobenius) gradient norm of* (22) *is*

$$\|\nabla_{\boldsymbol{Z}_{\mathrm{s}}^{\mathrm{cls}}} \widehat{\mathcal{L}}_{\mathrm{SimDINO}}\|_F \leq \frac{1}{\sqrt{M}} + \gamma \cdot \frac{\sqrt{d \min\{d, B\}/B}}{2\varepsilon}. \tag{23}$$

*Proof.* For the first term, we call upon Lemma C.4 which says that

$$\left\| \nabla_{(\boldsymbol{Z}_{\mathrm{s}}^{\mathrm{cls}})_b} \left\{ \frac{1}{2} \sum_{i=1}^{M_{\mathrm{glo}}} \sum_{j=1}^{M} \|(\boldsymbol{Z}_{\mathrm{t}}^{\mathrm{cls}})_{bi} - (\boldsymbol{Z}_{\mathrm{s}}^{\mathrm{cls}})_{bj}\|_2^2 \right\} \right\|_F \leq M_{\mathrm{glo}}\sqrt{M}. \tag{24}$$

This allows us to compute the gradient norm of the first term in (22) as

$$\left\| \nabla_{\boldsymbol{Z}_{\mathrm{s}}^{\mathrm{cls}}} \left\{ \frac{1}{2BMM_{\mathrm{glo}}} \sum_{b=1}^{B} \sum_{i=1}^{M_{\mathrm{glo}}} \sum_{j=1}^{M} \|(\boldsymbol{Z}_{\mathrm{t}}^{\mathrm{cls}})_{bi} - (\boldsymbol{Z}_{\mathrm{s}}^{\mathrm{cls}})_{bj}\|_2^2 \right\} \right\|_F \tag{25}$$

$$= \frac{1}{BMM_{\mathrm{glo}}} \left\| \nabla_{\boldsymbol{Z}_{\mathrm{s}}^{\mathrm{cls}}} \left\{ \frac{1}{2} \sum_{b=1}^{B} \sum_{i=1}^{M_{\mathrm{glo}}} \sum_{j=1}^{M} \|(\boldsymbol{Z}_{\mathrm{t}}^{\mathrm{cls}})_{bi} - (\boldsymbol{Z}_{\mathrm{s}}^{\mathrm{cls}})_{bj}\|_2^2 \right\} \right\|_F \tag{26}$$

$$\leq \frac{1}{BMM_{\mathrm{glo}}} \sum_{b=1}^{B} \left\| \nabla_{\boldsymbol{Z}_{\mathrm{s}}^{\mathrm{cls}}} \left\{ \frac{1}{2} \sum_{i=1}^{M_{\mathrm{glo}}} \sum_{j=1}^{M} \|(\boldsymbol{Z}_{\mathrm{t}}^{\mathrm{cls}})_{bi} - (\boldsymbol{Z}_{\mathrm{s}}^{\mathrm{cls}})_{bj}\|_2^2 \right\} \right\|_F \tag{27}$$

$$= \frac{1}{BMM_{\mathrm{glo}}} \sum_{b=1}^{B} \sqrt{\sum_{b'=1}^{B} \left\| \nabla_{(\boldsymbol{Z}_{\mathrm{s}}^{\mathrm{cls}})_{b'}} \left\{ \frac{1}{2} \sum_{i=1}^{M_{\mathrm{glo}}} \sum_{j=1}^{M} \|(\boldsymbol{Z}_{\mathrm{t}}^{\mathrm{cls}})_{bi} - (\boldsymbol{Z}_{\mathrm{s}}^{\mathrm{cls}})_{bj}\|_2^2 \right\} \right\|_F^2} \tag{28}$$

$$= \frac{1}{BMM_{\mathrm{glo}}} \sum_{b=1}^{B} \left\| \nabla_{(\boldsymbol{Z}_{\mathrm{s}}^{\mathrm{cls}})_b} \left\{ \frac{1}{2} \sum_{i=1}^{M_{\mathrm{glo}}} \sum_{j=1}^{M} \|(\boldsymbol{Z}_{\mathrm{t}}^{\mathrm{cls}})_{bi} - (\boldsymbol{Z}_{\mathrm{s}}^{\mathrm{cls}})_{bj}\|_2^2 \right\} \right\|_F \tag{29}$$

$$\leq \frac{1}{BMM_{\mathrm{glo}}} \sum_{b=1}^{B} M_{\mathrm{glo}}\sqrt{M} \tag{30}$$

$$= \frac{1}{\sqrt{M}}. \tag{31}$$

For the second term, we call upon Lemma C.6 which says that

$$\left\| \nabla_{(\boldsymbol{Z}_{\mathrm{s}}^{\mathrm{cls}})_{:,i}} R_\varepsilon\left(\frac{(\boldsymbol{Z}_{\mathrm{s}}^{\mathrm{cls}})_{:,i}^\top (\boldsymbol{Z}_{\mathrm{s}}^{\mathrm{cls}})_{:,i}}{B}\right) \right\|_F \leq \frac{\sqrt{d \min\{d, B\}/B}}{4\varepsilon}. \tag{32}$$

This allows us to compute the gradient norm of the second term in (22) as

$$
\left\| \nabla_{\boldsymbol{Z}_{\mathrm{s}}^{\mathrm{cls}}} \left\{ \frac{\gamma}{M_{\mathrm{glo}}} \sum_{i=1}^{M_{\mathrm{glo}}} R_{\varepsilon}\left( \frac{(\boldsymbol{Z}_{\mathrm{s}}^{\mathrm{cls}})_{:,i}^{\top}(\boldsymbol{Z}_{\mathrm{s}}^{\mathrm{cls}})_{:,i}}{B} \right) \right\} \right\|_F = \frac{\gamma}{M_{\mathrm{glo}}} \left\| \nabla_{\boldsymbol{Z}_{\mathrm{s}}^{\mathrm{cls}}} \left\{ \sum_{i=1}^{M_{\mathrm{glo}}} R_{\varepsilon}\left( \frac{(\boldsymbol{Z}_{\mathrm{s}}^{\mathrm{cls}})_{:,i}^{\top}(\boldsymbol{Z}_{\mathrm{s}}^{\mathrm{cls}})_{:,i}}{B} \right) \right\} \right\|_F \tag{33}
$$

$$
\leq \frac{\gamma}{M_{\mathrm{glo}}} \sum_{i=1}^{M_{\mathrm{glo}}} \left\| \nabla_{\boldsymbol{Z}_{\mathrm{s}}^{\mathrm{cls}}} R_{\varepsilon}\left( \frac{(\boldsymbol{Z}_{\mathrm{s}}^{\mathrm{cls}})_{:,i}^{\top}(\boldsymbol{Z}_{\mathrm{s}}^{\mathrm{cls}})_{:,i}}{B} \right) \right\|_F \tag{34}
$$

$$
= \frac{\gamma}{M_{\mathrm{glo}}} \sum_{i=1}^{M_{\mathrm{glo}}} \sqrt{\sum_{i'=1}^{M_{\mathrm{glo}}} \left\| \nabla_{(\boldsymbol{Z}_{\mathrm{s}}^{\mathrm{cls}})_{:,i'}} R_{\varepsilon}\left( \frac{(\boldsymbol{Z}_{\mathrm{s}}^{\mathrm{cls}})_{:,i}^{\top}(\boldsymbol{Z}_{\mathrm{s}}^{\mathrm{cls}})_{:,i}}{B} \right) \right\|_F^2} \tag{35}
$$

$$
= \frac{\gamma}{M_{\mathrm{glo}}} \sum_{i=1}^{M_{\mathrm{glo}}} \left\| \nabla_{(\boldsymbol{Z}_{\mathrm{s}}^{\mathrm{cls}})_{:,i}} R_{\varepsilon}\left( \frac{(\boldsymbol{Z}_{\mathrm{s}}^{\mathrm{cls}})_{:,i}^{\top}(\boldsymbol{Z}_{\mathrm{s}}^{\mathrm{cls}})_{:,i}}{B} \right) \right\|_F \tag{36}
$$

$$
\leq \frac{\gamma}{M_{\mathrm{glo}}} \sum_{i=1}^{M_{\mathrm{glo}}} \frac{1}{2\varepsilon} \sqrt{\frac{d \min\{d, B\}}{B}} \tag{37}
$$

$$
= \frac{\gamma}{2\varepsilon} \sqrt{\frac{d \min\{d, B\}}{B}}. \tag{38}
$$

Putting these two terms together via triangle inequality, we have that

$$
\|\nabla_{\boldsymbol{Z}_{\mathrm{s}}^{\mathrm{cls}}} \widehat{\mathcal{L}}_{\mathrm{SimDINO}}\|_F \leq \frac{1}{\sqrt{M}} + \frac{\gamma}{2\varepsilon} \sqrt{\frac{d \min\{d, B\}}{B}}, \tag{39}
$$

as desired. □

We can also come up with another result for SimDINOv2. This requires a slightly revised pipeline, as well as a different loss. For a given optimization step, we:

- Sample a minibatch $\{\boldsymbol{X}_1, \ldots, \boldsymbol{X}_B\}$.

- For each sample $\boldsymbol{X}_b$, draw $M_{\mathrm{glo}}$ global views $v_{\mathrm{glo}}^i$ and $M_{\mathrm{loc}}$ local views $v_{\mathrm{loc}}^j$. Define $\boldsymbol{X}_{b,\mathrm{glo}}^i := v_{\mathrm{glo}}^i(\boldsymbol{X}_b)$ and $\boldsymbol{X}_{b,\mathrm{loc}}^j := v_{\mathrm{loc}}^j(\boldsymbol{X}_b)$.

- For each sample $\boldsymbol{X}_b$, draw $M_{\mathrm{glo}}$ masks $m_b^i$ and apply them to each global view. Define $\boldsymbol{X}_{b,\mathrm{glo,mask}}^i := m_b^i(\boldsymbol{X}_{b,\mathrm{glo}}^i)$, and define $\mathbf{1}_{b,k}^i$ be the indicator variable of whether patch $k$ is masked by $m_b^i$.

- Compute the features $\boldsymbol{z}_{\mathrm{s}}^{\mathrm{cls}}(\boldsymbol{X}_{b,\mathrm{glo,mask}}^i)$, $\boldsymbol{Z}_{\mathrm{s}}^{\mathrm{patch}}(\boldsymbol{X}_{b,\mathrm{glo,mask}}^i)$, $\boldsymbol{z}_{\mathrm{s}}^{\mathrm{cls}}(\boldsymbol{X}_{b,\mathrm{loc}}^i)$, $\boldsymbol{z}_{\mathrm{t}}^{\mathrm{cls}}(\boldsymbol{X}_{b,\mathrm{glo}}^i)$, and $\boldsymbol{Z}_{\mathrm{t}}^{\mathrm{patch}}(\boldsymbol{X}_{b,\mathrm{glo}}^i)$.

- Form the tensors $\boldsymbol{Z}_{\mathrm{s}}^{\mathrm{cls}} \in \mathbb{R}^{B \times M \times d}$, $\boldsymbol{Z}_{\mathrm{t}}^{\mathrm{cls}} \in \mathbb{R}^{B \times M_{\mathrm{glo}} \times d}$, $\boldsymbol{Z}_{\mathrm{s,glo}}^{\mathrm{patch}} \in \mathbb{R}^{B \times M_{\mathrm{glo}} \times N_{\mathrm{glo}} \times d}$, $\boldsymbol{Z}_{\mathrm{t}}^{\mathrm{patch}} \in \mathbb{R}^{B \times M_{\mathrm{glo}} \times N_{\mathrm{glo}} \times d}$, where $M := M_{\mathrm{glo}} + M_{\mathrm{loc}}$, by

$$
(\boldsymbol{Z}_{\mathrm{s}}^{\mathrm{cls}})_{bi} = \begin{cases} \boldsymbol{z}_{\mathrm{s}}^{\mathrm{cls}}(\boldsymbol{X}_{b,\mathrm{glo,mask}}^i), & \text{if } 1 \leq i \leq M_{\mathrm{glo}}, \\ \boldsymbol{z}_{\mathrm{s}}^{\mathrm{cls}}(\boldsymbol{X}_{b,\mathrm{loc}}^{i-M_{\mathrm{glo}}}), & \text{otherwise} \end{cases}, \qquad (\boldsymbol{Z}_{\mathrm{t}}^{\mathrm{cls}})_{bi} = \boldsymbol{z}_{\mathrm{t}}^{\mathrm{cls}}(\boldsymbol{X}_{b,\mathrm{glo}}^i) \tag{40}
$$

$$
(\boldsymbol{Z}_{\mathrm{s,glo}}^{\mathrm{patch}})_{bi} = \boldsymbol{Z}_{\mathrm{s}}^{\mathrm{patch}}(\boldsymbol{X}_{b,\mathrm{glo,mask}}^i), \qquad (\boldsymbol{Z}_{\mathrm{t}}^{\mathrm{patch}})_{bi} = \boldsymbol{Z}_{\mathrm{t}}^{\mathrm{patch}}(\boldsymbol{X}_{b,\mathrm{glo}}^i). \tag{41}
$$

- Compute the loss

$$
\widehat{\mathcal{L}}_{\mathrm{SimDINOv2}} = \frac{1}{2} \left\{ \frac{1}{2BMM_{\mathrm{glo}}} \sum_{b=1}^{B} \sum_{i=1}^{M_{\mathrm{glo}}} \sum_{j=1}^{M} \|(\boldsymbol{Z}_{\mathrm{t}}^{\mathrm{cls}})_{bi} - (\boldsymbol{Z}_{\mathrm{s}}^{\mathrm{cls}})_{bj}\|_2^2 \right. \tag{42}
$$

$$
\left. + \frac{1}{2BM_{\mathrm{glo}}N_{\mathrm{glo}}} \sum_{b=1}^{B} \sum_{i=1}^{M_{\mathrm{glo}}} \sum_{n=1}^{N_{\mathrm{glo}}} \|(\boldsymbol{Z}_{\mathrm{t}}^{\mathrm{patch}})_{bin} - (\boldsymbol{Z}_{\mathrm{s,glo}}^{\mathrm{patch}})_{bin}\|_2^2 \mathbf{1}_{b,n}^i \right\}
$$

$$+ \frac{\gamma}{M_{\text{glo}}} \sum_{i=1}^{M_{\text{glo}}} R_\varepsilon \left( \frac{(\boldsymbol{Z}_{\text{s}}^{\text{cls}})_{:,i}^\top (\boldsymbol{Z}_{\text{s}}^{\text{cls}})_{:,i}}{B} \right),$$

where for $1 \leq i \leq M_{\text{glo}}$ we have $(\boldsymbol{Z}_{\text{s}}^{\text{cls}})_{:,i} = [(\boldsymbol{Z}_{\text{s}}^{\text{cls}})_{1,i}, \ldots, (\boldsymbol{Z}_{\text{s}}^{\text{cls}})_{B,i}]^\top \in \mathbb{R}^{B \times d}$.

Our main theorem for SimDINOv2 is the following:

**Theorem C.2.** *The (Frobenius) gradient norm of* (42) *is*

$$\|\nabla_{\boldsymbol{Z}_{\text{s}}^{\text{cls}}} \widehat{\mathcal{L}}_{\text{SimDINOv2}}\|_F \leq \frac{1}{2} \left( \frac{1}{\sqrt{M}} + \sqrt{\frac{1-\alpha}{N_{\text{glo}}}} \right) + \frac{\gamma}{2\varepsilon} \sqrt{\frac{d \min\{d, B\}}{B}}. \tag{43}$$

*where (recall) $\alpha \in [0, 1]$ is the fraction of patches of the input global view which is masked out.*

*Proof.* By the proof of Theorem C.1, the gradient norm of the first and third terms are known. So we study the second term. We can invoke Lemma C.5, which tells us that for every $b$ and $i$ we have

$$\left\| \nabla_{(\boldsymbol{Z}_{\text{s,glo}}^{\text{patch}})_{bi}} \left\{ \frac{1}{2} \sum_{n=1}^{N_{\text{glo}}} \|(\boldsymbol{Z}_{\text{t}}^{\text{patch}})_{bin} - (\boldsymbol{Z}_{\text{s,glo}}^{\text{patch}})_{bin}\|_2^2 \mathbf{1}_{b,n}^i \right\} \right\|_F = \sqrt{(1-\alpha)N_{\text{glo}}} \tag{44}$$

using the identity that for each $(i, b)$ pair that

$$\sum_{n=1}^{N_{\text{glo}}} \mathbf{1}_{b,n}^i = (1-\alpha)N_{\text{glo}}. \tag{45}$$

Then, it holds overall that

$$\left\| \nabla_{\boldsymbol{Z}_{\text{s}}} \left\{ \frac{1}{2BM_{\text{glo}}N_{\text{glo}}} \sum_{b=1}^{B} \sum_{i=1}^{M_{\text{glo}}} \sum_{n=1}^{N_{\text{glo}}} \|(\boldsymbol{Z}_{\text{t}}^{\text{patch}})_{bin} - (\boldsymbol{Z}_{\text{s,glo}}^{\text{patch}})_{bin}\|_2^2 \mathbf{1}_{b,n}^i \right\} \right\|_F \tag{46}$$

$$= \frac{1}{BM_{\text{glo}}N_{\text{glo}}} \left\| \nabla_{\boldsymbol{Z}_{\text{s}}} \left\{ \frac{1}{2} \sum_{b=1}^{B} \sum_{i=1}^{M_{\text{glo}}} \sum_{n=1}^{N_{\text{glo}}} \|(\boldsymbol{Z}_{\text{t}}^{\text{patch}})_{bin} - (\boldsymbol{Z}_{\text{s,glo}}^{\text{patch}})_{bin}\|_2^2 \mathbf{1}_{b,n}^i \right\} \right\|_F \tag{47}$$

$$\leq \frac{1}{BM_{\text{glo}}N_{\text{glo}}} \sum_{b=1}^{B} \sum_{i=1}^{M_{\text{glo}}} \left\| \nabla_{\boldsymbol{Z}_{\text{s}}} \left\{ \frac{1}{2} \sum_{n=1}^{N_{\text{glo}}} \|(\boldsymbol{Z}_{\text{t}}^{\text{patch}})_{bin} - (\boldsymbol{Z}_{\text{s,glo}}^{\text{patch}})_{bin}\|_2^2 \mathbf{1}_{b,n}^i \right\} \right\|_F \tag{48}$$

$$= \frac{1}{BM_{\text{glo}}N_{\text{glo}}} \sum_{b=1}^{B} \sum_{i=1}^{M_{\text{glo}}} \sqrt{\sum_{b'=1}^{B} \sum_{i'=1}^{M_{\text{glo}}} \left\| \nabla_{(\boldsymbol{Z}_{\text{s}}^{\text{patch}})_{b'i'}} \left\{ \frac{1}{2} \sum_{n=1}^{N_{\text{glo}}} \|(\boldsymbol{Z}_{\text{t}}^{\text{patch}})_{bin} - (\boldsymbol{Z}_{\text{s,glo}}^{\text{patch}})_{bin}\|_2^2 \mathbf{1}_{b,n}^i \right\} \right\|_F^2} \tag{49}$$

$$= \frac{1}{BM_{\text{glo}}N_{\text{glo}}} \sum_{b=1}^{B} \sum_{i=1}^{M_{\text{glo}}} \left\| \nabla_{(\boldsymbol{Z}_{\text{s}}^{\text{patch}})_{bi}} \left\{ \frac{1}{2} \sum_{n=1}^{N_{\text{glo}}} \|(\boldsymbol{Z}_{\text{t}}^{\text{patch}})_{bin} - (\boldsymbol{Z}_{\text{s,glo}}^{\text{patch}})_{bin}\|_2^2 \mathbf{1}_{b,n}^i \right\} \right\|_F \tag{50}$$

$$= \frac{1}{BM_{\text{glo}}N_{\text{glo}}} \sum_{b=1}^{B} \sum_{i=1}^{M_{\text{glo}}} \sqrt{(1-\alpha)N_{\text{glo}}} \tag{51}$$

$$= \frac{\sqrt{1-\alpha}}{\sqrt{N_{\text{glo}}}}. \tag{52}$$

Using the gradients computed in Theorem C.1 and the triangle inequality, it holds that

$$\|\nabla_{\boldsymbol{Z}_{\text{s}}} \widehat{\mathcal{L}}_{\text{SimDINOv2}}\|_F \leq \frac{1}{2} \left( \frac{1}{\sqrt{M}} + \sqrt{\frac{1-\alpha}{N_{\text{glo}}}} \right) + \frac{\gamma}{2\varepsilon} \sqrt{\frac{d \min\{d, B\}}{B}} \tag{53}$$

as desired. □

*Remark* C.3. In the main body, in order to obtain a prescription for how to scale $\gamma$ with the batch size $B$, we choose $\gamma$ to make the different terms in (22) and (42) have equal magnitude. Given the explicit rate in Theorem C.1, the value of $\gamma$ chosen in SimDINO is

$$\gamma_{\text{SimDINO}} = 2\varepsilon \sqrt{\frac{B}{dM \min\{d, B\}}}. \tag{54}$$

Meanwhile using Theorem C.2, the value of $\gamma$ chosen in SimDINOv2 is

$$\gamma_{\text{SimDINOv2}} = \varepsilon \left( \frac{1}{\sqrt{M}} + \sqrt{\frac{1-\alpha}{N_{\text{glo}}}} \right) \sqrt{\frac{B}{d \min\{d, B\}}}. \tag{55}$$

If, for instance, we are just interested in scaling $\gamma$ with the batch size $B$ (so that $\varepsilon, \alpha, M, N_{\text{glo}}, d$ are held constant), then $\gamma_{\text{SimDINO}}$ and $\gamma_{\text{SimDINOv2}}$ have the same asymptotic order. In practice, we take these prescriptions for $\gamma$ up to a multiplicative constant, which is tuned on a single setting and can then be transferred to different settings.

### C.1. Auxiliary Lemmas

**Lemma C.4** (Scale of Gradient of Pairwise Distance Term). *Let $d, m, n$ be positive integers. Let $\boldsymbol{A} \in \mathbb{R}^{m \times d}$ and $\boldsymbol{B} \in \mathbb{R}^{n \times d}$ have rows $\boldsymbol{a}_i$ and $\boldsymbol{b}_j$ which are unit-$\ell^2$-norm, i.e., $\|\boldsymbol{a}_i\|_2 = \|\boldsymbol{b}_j\|_2 = 1$ for $i \in \{1, \ldots, m\}, j \in \{1, \ldots, n\}$. Then*

$$\left\| \nabla_{\boldsymbol{B}} \left\{ \frac{1}{2} \sum_{i=1}^m \sum_{j=1}^n \|\boldsymbol{a}_i - \boldsymbol{b}_j\|_2^2 \right\} \right\|_F \leq m\sqrt{n}. \tag{56}$$

*Proof.* Since all rows are normalized,

$$\frac{1}{2} \sum_{i=1}^m \sum_{j=1}^n \|\boldsymbol{a}_i - \boldsymbol{b}_j\|_2^2 = -\sum_{i=1}^m \sum_{j=1}^n \boldsymbol{a}_i^\top \boldsymbol{b}_j = -\left[ \sum_{i=1}^m \boldsymbol{a}_i \right]^\top \left[ \sum_{j=1}^n \boldsymbol{b}_j \right] \tag{57}$$

$$= -(\mathbf{1}_m^\top \boldsymbol{A})(\mathbf{1}_n^\top \boldsymbol{B})^\top \tag{58}$$

$$= -\mathbf{1}_m^\top \boldsymbol{A} \boldsymbol{B}^\top \mathbf{1}_n. \tag{59}$$

A matrix calculus computation shows that

$$\nabla_{\boldsymbol{B}} \left[ -\mathbf{1}_m^\top \boldsymbol{A} \boldsymbol{B}^\top \mathbf{1}_n \right] = -\mathbf{1}_n \mathbf{1}_m^\top \boldsymbol{A}. \tag{60}$$

To bound the norm of this term, we have

$$\| -\mathbf{1}_n \mathbf{1}_m^\top \boldsymbol{A} \|_F = \|\mathbf{1}_n \mathbf{1}_m^\top \boldsymbol{A}\|_F \leq \|\mathbf{1}_n \mathbf{1}_m^\top\|_{\text{op}} \|\boldsymbol{A}\|_F. \tag{61}$$

It is easy to show by matrix algebra that

$$\|\mathbf{1}_n \mathbf{1}_m^\top\|_{\text{op}} = \sqrt{mn}, \tag{62}$$

and that, since the rows of $\boldsymbol{A}$ are normalized,

$$\|\boldsymbol{A}\|_F = \sqrt{\sum_{i=1}^m \|\boldsymbol{a}_i\|_2^2} = \sqrt{m \cdot 1} = \sqrt{m}. \tag{63}$$

Putting these together it holds that

$$\left\| \nabla_{\boldsymbol{B}} \left\{ \frac{1}{2} \sum_{i=1}^m \sum_{j=1}^n \|\boldsymbol{a}_i - \boldsymbol{b}_j\|_2^2 \right\} \right\|_F \leq m\sqrt{n} \tag{64}$$

as desired. $\square$

**Lemma C.5** (Scale of Gradient of Patchwise Distance Term). *Let $d, n$ be positive integers. Let $\boldsymbol{A}, \boldsymbol{B} \in \mathbb{R}^{n \times d}$ have rows $\boldsymbol{a}_i$ and $\boldsymbol{b}_j$ which are unit-$\ell^2$-norm, i.e., $\|\boldsymbol{a}_i\|_2 = \|\boldsymbol{b}_j\|_2 = 1$ for $i, j \in \{1, \ldots, n\}$. Then*

$$\left\| \nabla_{\boldsymbol{B}} \left\{ \frac{1}{2} \sum_{i=1}^{n} \|\boldsymbol{a}_i - \boldsymbol{b}_i\|_2^2 \right\} \right\|_F = \sqrt{n}. \tag{65}$$

*Proof.* Since all rows are normalized,

$$\frac{1}{2} \sum_{i=1}^{n} \|\boldsymbol{a}_i - \boldsymbol{b}_i\|_2^2 = -\sum_{i=1}^{n} \boldsymbol{a}_i^\top \boldsymbol{b}_i = -\sum_{i=1}^{n} \sum_{j=1}^{d} (\boldsymbol{A})_{ij} (\boldsymbol{B})_{ij} = -\operatorname{tr}(\boldsymbol{A} \boldsymbol{B}^\top). \tag{66}$$

A matrix calculus computation shows that

$$\nabla_{\boldsymbol{B}} [-\operatorname{tr}(\boldsymbol{A} \boldsymbol{B}^\top)] = \boldsymbol{A}. \tag{67}$$

The Frobenius norm of this term can be explicitly calculated as

$$\|\boldsymbol{A}\|_F = \sqrt{\sum_{i=1}^{n} \|\boldsymbol{a}_i\|_2^2} = \sqrt{n \cdot 1} = \sqrt{n}. \tag{68}$$

Putting this together, we obtain

$$\left\| \nabla_{\boldsymbol{B}} \left\{ \frac{1}{2} \sum_{i=1}^{n} \|\boldsymbol{a}_i - \boldsymbol{b}_i\|_2^2 \right\} \right\|_F = \sqrt{n}, \tag{69}$$

as desired. $\qquad\square$

**Lemma C.6** (Scale of Gradient of Coding Rate Term). *Let $d, n$ be positive integers. We have*

$$\max_{\substack{\boldsymbol{Z} \in \mathbb{R}^{n \times d} \\ \|\boldsymbol{z}_i\|_2 = 1 \; \forall i}} \left\| \nabla_{\boldsymbol{Z}} R_\varepsilon \left( \frac{\boldsymbol{Z}^\top \boldsymbol{Z}}{n} \right) \right\|_F \leq \frac{1}{2\varepsilon} \sqrt{\frac{d \min\{d, n\}}{n}} \tag{70}$$

*where $\boldsymbol{z}_i$ is the $i$th row of $\boldsymbol{Z}$.*

*Proof.* Let $\alpha := d/(n\varepsilon^2)$ and let $f \colon \mathbb{R}^{n \times d} \to \mathbb{R}$ be defined by

$$f(\boldsymbol{Z}) := \frac{1}{2} \operatorname{logdet}(\boldsymbol{I} + \alpha \boldsymbol{Z}^\top \boldsymbol{Z}), \tag{71}$$

i.e., $f(\boldsymbol{Z}) = R_\varepsilon(\boldsymbol{Z}^\top \boldsymbol{Z}/n)$. Now, let $r := \min\{d, n\}$. For any matrix $\boldsymbol{M}$, let $\sigma_i(\boldsymbol{M})$ be its $i^{\text{th}}$ largest singular value, for $i = 1, \ldots, d$. First, note that since $\|\boldsymbol{z}_i\|_2 = 1$ for all $i$, it holds

$$\sum_{i=1}^{r} \sigma_i(\boldsymbol{Z})^2 = \sum_{i=1}^{d} \sigma_i(\boldsymbol{Z})^2 = \sum_{i=1}^{d} \sigma_i(\boldsymbol{Z}^\top \boldsymbol{Z}) = \operatorname{tr}(\boldsymbol{Z}^\top \boldsymbol{Z}) = \sum_{i=1}^{d} (\boldsymbol{Z}^\top \boldsymbol{Z})_{ii} = \sum_{i=1}^{d} \underbrace{\|\boldsymbol{z}_i\|_2}_{=1} = d. \tag{72}$$

Now, we can simplify the gradient. It holds

$$\nabla f(\boldsymbol{Z}) = \alpha \boldsymbol{Z} (\boldsymbol{I} + \alpha \boldsymbol{Z}^\top \boldsymbol{Z})^{-1}. \tag{73}$$

Thus, it holds that

$$\|\nabla f(\boldsymbol{Z})\|_F^2 = \operatorname{tr}([\nabla f(\boldsymbol{Z})][\nabla f(\boldsymbol{Z})]^\top) \tag{74}$$

$$= \alpha^2 \operatorname{tr}(\boldsymbol{Z}(\boldsymbol{I} + \alpha \boldsymbol{Z}^\top \boldsymbol{Z})^{-2} \boldsymbol{Z}^\top). \tag{75}$$

Using that the trace is the sum of singular values, it holds by taking the SVD of $\boldsymbol{Z}$ that

$$\operatorname{tr}(\boldsymbol{Z}(\boldsymbol{I} + \alpha \boldsymbol{Z}^\top \boldsymbol{Z})^{-2} \boldsymbol{Z}^\top) = \sum_{i=1}^r \sigma_i(\boldsymbol{Z}(\boldsymbol{I} + \alpha \boldsymbol{Z}^\top \boldsymbol{Z})^{-2} \boldsymbol{Z}^\top) \tag{76}$$

$$= \sum_{i=1}^r \frac{\sigma_i(\boldsymbol{Z})^2}{[1 + \alpha \sigma_i(\boldsymbol{Z})^2]^2}. \tag{77}$$

In this case we directly optimize over the singular values, obtaining the problem

$$\max_{\substack{\boldsymbol{Z} \in \mathbb{R}^{n \times d} \\ \|\boldsymbol{z}_i\|_2 = 1 \ \forall i}} \|\nabla f(\boldsymbol{Z})\|_F \leq \max_{\substack{\boldsymbol{x} \in \mathbb{R}^r \\ x_i \geq 0 \ \forall i \\ \sum_{i=1}^r x_i = d}} \sum_{i=1}^r \frac{x_i}{(1 + \alpha x_i)^2}. \tag{78}$$

The function $t \mapsto \frac{t}{(1 + \alpha t)^2}$ on $[0, \infty)$ has a global maximum at $t = \frac{1}{\alpha}$, and the value is $\frac{1}{4\alpha}$. Therefore it follows that

$$\max_{\substack{\boldsymbol{x} \in \mathbb{R}^r \\ x_i \geq 0 \ \forall i \\ \sum_{i=1}^r x_i = d}} \sum_{i=1}^r \frac{x_i}{(1 + \alpha x_i)^2} \leq \max_{\substack{\boldsymbol{x} \in \mathbb{R}^r \\ x_i \geq 0 \ \forall i}} \sum_{i=1}^r \frac{x_i}{(1 + \alpha x_i)^2} = \frac{r}{4\alpha}. \tag{79}$$

Unpacking this notation, we obtain

$$\|\nabla f(\boldsymbol{Z})\|_F^2 \leq \alpha^2 \cdot \frac{r}{4\alpha} = \frac{\alpha r}{4} = \frac{d \min\{d, n\}}{4n\varepsilon^2}. \tag{80}$$

Taking square roots, it holds

$$\|\nabla f(\boldsymbol{Z})\|_F \leq \frac{1}{2\varepsilon} \sqrt{\frac{d \min\{d, n\}}{n}}. \tag{81}$$

Therefore,

$$\left\| \nabla_{\boldsymbol{Z}} R_\varepsilon \left( \frac{\boldsymbol{Z}^\top \boldsymbol{Z}}{n} \right) \right\|_F = \|\nabla f(\boldsymbol{Z})\|_F \leq \frac{1}{2\varepsilon} \sqrt{\frac{d \min\{d, n\}}{n}} \tag{82}$$

as desired. $\square$

*Remark* C.7. It is possible that the inequality

$$\max_{\substack{\boldsymbol{Z} \in \mathbb{R}^{n \times d} \\ \|\boldsymbol{z}_i\|_2 = 1 \ \forall i}} \|\nabla f(\boldsymbol{Z})\|_F \leq \max_{\substack{\boldsymbol{x} \in \mathbb{R}^r \\ x_i \geq 0 \ \forall i \\ \sum_{i=1}^r x_i = d}} \sum_{i=1}^r \frac{x_i}{(1 + \alpha x_i)^2}. \tag{83}$$

is met with equality; proving this would require exhibiting a $\boldsymbol{Z}$ fulfilling the constraints of the first problem such that it has the prescribed singular values which solve the second problem. We do not need to do so here for the purposes of using the bound (e.g., for learning rate scaling).

*Remark* C.8. While the quick-and-dirty bound

$$\max_{\substack{\boldsymbol{x} \in \mathbb{R}^r \\ x_i \geq 0 \ \forall i \\ \sum_{i=1}^r x_i = d}} \sum_{i=1}^r \frac{x_i}{(1 + \alpha x_i)^2} \leq \frac{r}{4\alpha}, \tag{84}$$

by way of ignoring the constraint $\sum_{i=1}^r x_i = d$ seems like it could significantly loosen the bound, we do not believe this is the case. In particular, when $1/\alpha \leq d/r$, note that setting $x_1 = \cdots = x_{r-1} = 1/\alpha$ and $x_r = d - (r-1)/\alpha$ sandwiches the objective between $(r-1)/(4\alpha)$ and $r/(4\alpha)$, so the maximum is at least the same asymptotic order, in the very reasonable case that $\varepsilon$ is small enough that $1/\alpha \leq d/r$, i.e., using the definition of $\alpha$, such that

$$\frac{1}{\alpha} \leq \frac{d}{r} \iff \varepsilon^2 \leq \frac{d^2}{n \min\{d, n\}} \iff \varepsilon^2 \leq \max\left\{ \frac{d}{n}, \frac{d^2}{n^2} \right\}. \tag{85}$$

Similar strategies should hold if we allow for an absolute constant $c \geq 1$ such that $1/\alpha \leq cd/r$, etc, relaxing the requirement while preserving the asymptotic order of the LHS of (84).

# D. Training Pipeline Pseudocode

In this section we provide pseudocode for the training pipelines of SimDINO and SimDINOv2.

---

**Algorithm 1** SimDINO training pipeline.

---

```
# fs, ft: student and teacher networks, this time outputting ONLY the cls token feature
# eps: coding rate regularization quantization hyperparameter
# gamma: coding rate regularization strength hyperparameter
# lam: teacher network EMA rate
ft.params = fs.params
for x in loader: # load a minibatch x of B samples
    xg, xl = global_views(x), local_views(x) # (B, M_glo, N_glo, D), (B, M_loc, N_loc, D)

    zsg, zsl = fs(xg), fs(xl) # student output (B, M_glo, d), (B, M_loc, d)
    ztg = ft(xg) # teacher output (B, M_glo, d)

    zs = cat([zsg, zsl], dim=1)  # (B, M, d) where M = M_loc + M_glo

    sq_dists = sum((zs.view(B,M,1,d) - ztg.view(B,1,M_glo,d)) ** 2, dim=3) # (B, M, M_glo)

    zsg_bdim = zsg.transpose(0, 1)  # (M_glo, B, d)
    covs = zsg_bdim.transpose(1, 2) @ zsg_bdim / B  # (M_glo, d, d)
    R_eps = batch_logdet(I_d.unsqueeze(0) + d/(eps**2) * covs)  # (M_glo)

    loss = mean(sq_dists) - gamma * mean(R_eps)
    loss.backward() # back-propagate

    # student and teacher updates
    update(fs) # SGD or Adam
    ft.params = lam * ft.params + (1 - lam) * fs.params
```

---

# E. Implementation Details

The training codes and hyperparameters for SimDINO and SimDINOv2 are derived from the released official settings in DINO and DINOv2 separately, see Table 4 for detailed comparison. Notes that for SimDINOv2, we choose to use bfloat16 dtype in student backbone parameters and reductions for better numerical stability while other modules uses the same FSDP mixed precision settings from DINOv2.

# F. Additional Experiments

### F.1. Ablations on Stability of DINO Training

In Table 6, we study the optimization behavior and stability of DINO by varying hyperparameters that are specific to its pipeline. Specifically, we select teacher momentum, whether to apply normalization for the last layer, and teacher temperature. We vary each of them and study their impact on DINO training. As shown in Table 6, moderate adjustments for each component leads to divergence (during early training stages). These results suggest DINO training can be highly unstable and requires careful tuning efforts.

### F.2. Ablation Studies on Batch Sizes

We vary the batch sizes when training ViT-S using SimDINO and report the results in Table 7. We observe that SimDINO is robust to the choice of batch sizes and can converge to reasonably good performance with a smaller batch size of 256.

### F.3. Experiments on Longer Training

More training epochs in SSL typically lead to better performance. We provide the performance of SimDINO when doubling the number of epochs in Table 8. Clearly, these results show the efficacy of longer training for SimDINO.

**Algorithm 2** SimDINOv2 training pipeline.

```
# fs, ft: student and teacher networks, this time outputting BOTH the cls token feature
    and patch token features
# eps: coding rate regularization quantization hyperparameter
# gamma: coding rate regularization strength hyperparameter
# lam: teacher network EMA rate
# alpha: proportion of patches that get masked
ft.params = fs.params
for x in loader: # load a minibatch x of B samples
    m = generate_mask(x, alpha)  # boolean mask (B, M_glo, N_glo)

    xg, xl = global_views(x), local_views(x) # (B, M_glo, N_glo, D), (B, M_loc, N_loc, D)

    xmg = apply_mask(xg, m) # (B, M_glo, N_glo, D)

    zsg, Zsg = fs(xmg)# student on masked global views (B, M_glo, d), (B, M_glo, N_glo, d)
    zsl, Zsl = fs(xl) # student output on local views (B, M_loc, d), (B, M_loc, N_loc, d)

    ztg, Ztg = ft(xg) # teacher output on global views (B, M_glo, d), (B, M_glo, N_glo, d)

    zs = cat([zsg, zsl], dim=1)  # (B, M, d), M = M_loc + M_glo

    sq_dists = sum((zs.view(B,M,1,d) - ztg.view(B,1,M_glo,d)) ** 2, dim=3) # (B, M, M_glo)
    patch_sq_dists = mean(sum((Zsg - Ztg) ** 2, dim=3) * m, dim=2)  # (B, M_glo)

    zsg_bdim = zsg.transpose(0, 1)  # (M_glo, B, d)
    covs = zsg_bdim.transpose(-2, -1) @ zsg_bdim / B  # (M_glo, d, d)
    R_eps = batch_logdet(I_d.unsqueeze(0) + d/(eps**2) * covs)  # (M_glo)

    loss = (mean(sq_dists) + mean(patch_sq_dists))/2 - gamma * mean(R_eps)
    loss.backward() # back-propagate

    # student and teacher updates
    update(fs) # SGD or Adam
    ft.params = lam * ft.params + (1 - lam) * fs.params
```

## F.4. DINO without Self-Distillation

Due to the explicit coding rate regularization, it is possible to train SimDINO without self-distillation. To validate this, we train ViT-S models on ImageNet-1k by setting the teacher network to be the student network at each iteration, effectively removing the EMA operation. Results are presented in Table 9. We can see that the original DINO collapses under this setup for reasons discussed in Appendix B, while SimDINO is able to yield non-trivial performance. It is worth noting that compared to training with full self-distillation, this variant primarily lags behind in terms of $k$-NN performance while the gap in linear probe is significantly smaller.

## F.5. Ablations on Loss Functions

To prevent representation collapse, we adopt the coding rate objective $R_\varepsilon$ in SimDINO and SimDINOv2. In this part, we examine the effectiveness of the coding rate formulation and compare it with other loss functions that aim to prevent collapse. Specifically, we swap the coding rate function with the following three widely-used objectives: (1) the vanilla contrastive loss $\ell_{\text{contrastive}}$ based on InfoNCE as in SimCLR (Chen et al., 2020), (2) the uniform loss $\ell_{\text{uniform}}$ proposed in (Wang & Isola, 2020) that encourages the representations to be uniform on the unit sphere, (3) the Barlow Twins loss $\ell_{\text{bt}}$ proposed in (Zbontar et al., 2021) that penalizes off-diagonal terms while promoting the on-diagonal terms on the covariance matrix of learned representations. Results are presented in Table 5. We observe that the coding rate objective consistently performs better than the other choices considered, validating our design.

## F.6. Visualization of Attention Maps

Following (Oquab et al., 2023; Caron et al., 2021), we provide visualizations of self-attention maps of different models for qualitative comparison. We use test images that do not appear during pretraining. More concretely, we compute and visualize the average of self-attention maps across all attention heads from the last layer in Figure 5. It is clear from the attention maps that all methods studied in our paper lead to prominent segmentation properties that emerge from vision self-supervised learning.

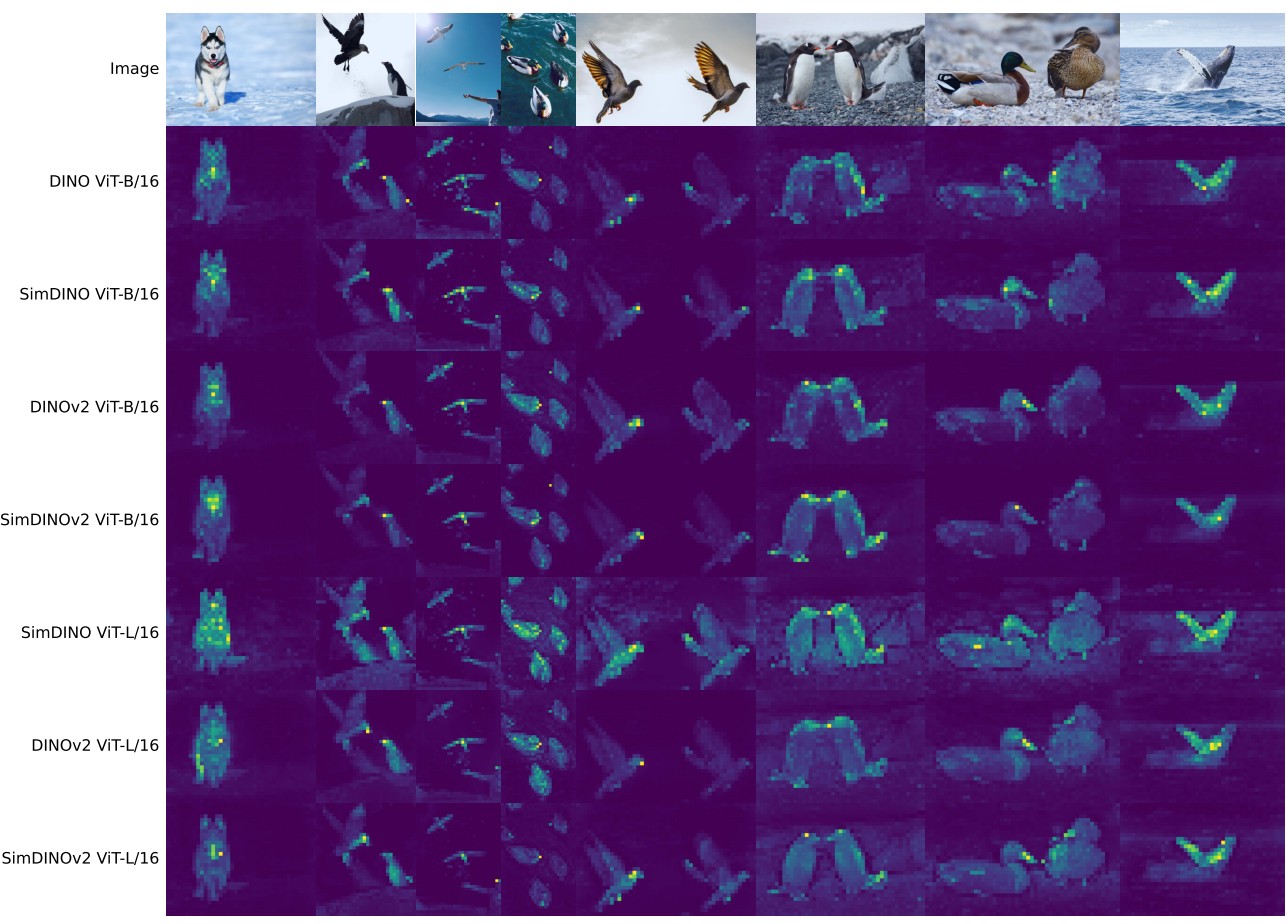

*Figure 5.* **Visualization of average self-attention maps** obtained from both DINO(v2) and SimDINO(v2) algorithms.

| | **Hyperparameter** | **SimDINOv2** | **DINOv2** | **SimDINO** | **DINO** |
|---|---|---|---|---|---|
| **Model** | Patch size | 16 | | | |
| | Register tokens | 4 | | 0 | |
| | Pos-embedding anti-alias | True | | False | |
| | Init layer scale | 0.1 | 1e-5 | - | |
| | Drop path rate | 0.3 | | 0.1 | |
| | Weight normalize last layer | removed | True | removed | True |
| | Output prototypes K | removed | 65536 | removed | 65536 |
| **Pipeline** | Init EMA momentum | 0.9 | 0.992 | 0.996 | |
| | Centering temperature | removed | 0.07 | removed | 0.07 |
| | Warm-up temperature | removed | 0.04 | removed | 0.04 |
| | Warm-up temperature epochs | removed | 30 | removed | 30 |
| | iBOT sample prob. | 0.5 | | - | |
| | iBOT mask ratio | 0.1-0.5 | | - | |
| | iBOT head untying | False | | - | |
| | Koleo loss weight | removed | 0.1 | - | |
| **Data** | Global crops scale | 0.4 - 1 | | | |
| | Local crops scale | 0.05 - 0.4 | | | |
| | Local crops number | 10 | | | |
| | Global crops size | 224 | | | |
| | Local crops size | 96 | | | |
| **Optim.** | Batch size | 128x8 | | 64x8 | |
| | Epochs | 100 | | | |
| | Warm-up epochs | 10 | | | |
| | Freeze last layer epochs | removed | 1 | removed | 1 |
| | Learning rate | 0.004 | | 0.002 | |
| | Layerwise lr decay | 0.9 | | - | |
| | Weight decay | 0.04 | | | |
| | Weight decay end | 0.4 | | | |
| | Gradient clip | 3.0 | | 0.3 | |

*Table 4.* Training hyperparameters used in the experiments

| Objectives to prevent collapse | $k$-NN | Linear |
|---|---|---|
| contrastive loss $\ell_{\text{contrastive}}$ | 79.4 | 81.0 |
| uniform loss $\ell_{\text{uniform}}$ | 77.2 | 82.1 |
| Barlow Twins loss $\ell_{\text{bt}}$ | - | - |
| coding rate loss $R_\varepsilon$ (ours) | 81.1 | 82.4 |

*Table 5.* **Ablations on Loss Functions**. We evaluate ViT-L pretrained on ImageNet-1k for 100 epochs by swapping the coding rate function in SimDINOv2 with other choices of collapse-prevention objectives. Barlow Twins loss causes representation collapse in our experiments.

*Table 6.* **Sensitivity of DINO on selected hyperparameters.** We pick three DINO-specific hyperparameters (i.e. teacher momentum, last-layer head normalization, teacher temperature) of the official configuration in (Caron et al., 2021) to study their impact. Varying each one leads to divergence in early training.

| Config | Mom. | Norm. | Temp. | $k$-NN |
|---|---|---|---|---|
| official (400ep) | 0.996 | ✓ | $0.04 \rightarrow 0.07$ | 76.1 |
| | 0.90 | ✓ | $0.04 \rightarrow 0.07$ | NaN |
| | 0.996 | ✗ | $0.04 \rightarrow 0.07$ | NaN |
| | 0.996 | ✓ | 0.07 | NaN |

| Batch size | 256 | 512 | 1024 |
|---|---|---|---|
| $k$-NN | 68.3 | 69.7 | 69.6 |

*Table 7.* **Effect of batch sizes**. We evaluate $k$-NN accuracy of ViT-S pretrained on ImageNet-1k for 100 epochs.

| Method | Epochs | $k$-NN | Linear |
|---|---|---|---|
| DINO | 100 | 72.9 | 76.3 |
| SimDINO | 100 | 74.9 | 77.3 |
| DINO | 200 | 73.6 | 77.1 |
| SimDINO | 200 | 76.0 | 77.7 |
| DINO* | 400 | 76.1 | 78.0 |

*Table 8.* **Effect of training epochs**. We evaluate ViT-B pretrained on ImageNet-1k for 100, 200, and 400 epochs. (DINO* 400 epochs evaluated on provided checkpoint )

| Method | Model | self-distillation | Epochs | $k$-NN | Linear |
|---|---|---|---|---|---|
| DINO | ViT-S | ✗ | 100 | – | – |
| SimDINO | ViT-S | ✗ | 100 | 58.6 | 68.0 |
| SimDINO | ViT-S | ✓ | 100 | 69.7 | 73.6 |

*Table 9.* Performance on ImageNet-1K without self-distillation.

