# OpenReview forum: "Simplifying DINO via Coding Rate Regularization"
_ICML.cc/2025/Conference — ICML 2025 poster_

### Official Review · Reviewer_ubYD · 2025-03-02

**Overall Recommendation:** 2

**Summary:**

The authors proposed the coding rate regularization technique to improve the training method of DINO and DINOv2, enhancing their performance.

**Claims And Evidence:**

2-1. The careful review of the original DINO and DINOv2 provides an intuitive and well-explained reasoning for why collapse occurs.

2-2. It is impressive that a method for effectively selecting the necessary hyperparameters for the proposed regularization is presented, along with a well-supported justification.

**Essential References Not Discussed:**

8-1. There do not appear to be any missing key references.

**Experimental Designs Or Analyses:**

5-1. The intention to improve upon DINO and DINOv2 is clear, but why is there no comparison with other self-supervised learning methods?

5-2. Is there a reason why few-shot learning performance, which is commonly evaluated in most representation learning papers, is not included?

**Methods And Evaluation Criteria:**

3-1. As stated in the title, it is clear that simplifying DINO is the main focus of the paper. However, the proposed coding rate regularization seems applicable to other representation learning methods that use contrastive learning. Are there any experimental results demonstrating this?

**Other Comments Or Suggestions:**

I have no further comments.

**Other Strengths And Weaknesses:**

10W-1. How does the actual regularization value change? Is there an analysis comparing cases where collapse occurs and where it does not?

10W-2. Is there a performance comparison when applying other regularization methods?

**Questions For Authors:**

Please refer to the comments provided in Questions 2-11.

**Relation To Broader Scientific Literature:**

7-1. Self-supervised representation learning is an actively researched field. While various methods have been proposed, representation collapse remains a critical issue that needs to be addressed. This work provides a simple yet effective solution to this problem.

**Theoretical Claims:**

4-1. The proof provided in the Appendix has been reviewed, and no issues were found.

---

> ### Author Rebuttal · Authors · 2025-04-01
>
> Dear Reviewer ubYD,
>
> We deeply appreciate your insightful review and am pleased to see that you find our work intuitive to follow, and well-justified by extensive empirical studies. Here we attempt to address the concerns you have raised in the review.
>
> 3-1: This is a great suggestion. It is possible to apply coding rate to other representation learning methods, as it is a principled measure of non-collapse (we provide a sketch of theoretical justification in our response for Reviewer 5gcy). As you rightfully point out, we focus on simplifying DINO and DINOv2 in this work as they are widely used and attain SOTA performance. The study of extending the coding rate to other learning frameworks is interesting, but out of scope for this paper and we leave it for future work.
>
> 5-1: Thanks for your comment. As discussed in the previous point, this work focuses on the DINO family of models as they are very widely used and typically best-performing across various downstream tasks. In ImageNet classification results, we also have included additional baselines (e.g. MoCov3) for reference. With that said, we are happy to incorporate more baselines in the final version of the paper.
>
> 5-2: We would like to respectfully note that few-shot learning is not really necessary here, since we already obtain linearly separable features from DINO/SimDINO training (demonstrated by the superior $k$-NN performance). Due to this property, few-shot learning was not evaluated in the original DINOv2 paper. We hope this helps clarify and resolve your concern.
>
> 10-W1: Thank you for your question. We can attach training plots in the final version of the paper. As long as the balancing coefficient $\gamma$ is selected appropriately, the coding rate value will steadily increase, meaning that the features are spread out without collapse. If it’s too small, then feature alignment loss will dominate and lead to collapse (coding rate approaches 0). If it’s too large, the opposite happens, where the coding rate dominates. In this case, the features don’t collapse but feature alignment between global and local views is no longer enforced. We will follow your suggestion and put these analyses in the paper.
>
> 10-W2: We appreciate your suggestion. In our preliminary test, using regularization such as direct contrastive loss leads to worse performance than coding rate regularization. We will run and include additional experiments with other regularization methods in the revised version of the paper.
>
> We are grateful to your suggestions and comments, which will no doubt improve the quality of our work. Please let us know if you have further questions or concerns, and we are happy to address them.

---

> > ### Comment · Reviewer_ubYD · 2025-04-03
> >
> > Thank you for taking the time to address the questions. While I appreciate the effort to clarify the contributions, I believe that my concerns regarding the proposed regularization—one of the core aspects of the paper—remain insufficiently addressed. As such, I will need to revise my initial score.

---

> > > ### Author Response · Authors · 2025-04-08
> > >
> > > We greatly appreciate your feedback, but we are not sure what aspects of your concerns regarding our proposed regularization are not adequately addressed. Nevertheless, we will try our best to further clarify and hopefully resolve your concerns.
> > >
> > > > Comparison with other regularization methods
> > >
> > > We greatly appreciate your suggestions and we have since tested on three different regularization methods to compare with our proposed coding rate regularization. The following results are also provided in our response to Reviewer nrJd. Concretely, we fix other settings to be the same as SimDINOv2 on ViT-L/16 and only vary the choice of the regularizer applied on the student features of global views for a fair comparison. We report their specific formulation and KNN performance as follows.
> > >
> > > - the vanilla contrastive objective (i.e. explicitly repelling negative samples and attracting positive ones) as in SimCLR[1]. Compared to our result, this setting results in performance decrease of about 2 points (79.4 vs 81.1).
> > > - the uniform loss proposed in [2] that encourages the representation to be uniform on the unit sphere. This settings leads to performance decrease of about 4 points (77.2 vs 81.1).
> > > - Barlow Twins loss proposed in [3] that penalizes off-diagonal terms while promoting the on-diagonal terms on the covariance matrix of learned representations. This setting causes NaN in our experiments so far. It might be related to the sensitivity of the coefficient within the loss (as it needs to balance the off-diagonal and on-diagonal terms) and we are still investigating.
> > >
> > > > Evolution of coding rate term value
> > >
> > > We hope our original response addresses this specific conern. As we said previously, we will put training plots that show the evolution of our loss function values during training in different settings in our camera-ready version. Also, we will be open-sourcing our code for reproduction of our results.
> > >
> > > > Application to other representation learning frameworks
> > >
> > > We greatly appreciate your advice. As said in our earlier responses, this work focuses on DINO and DINOv2 as they represent the state-of-the-art methods at the moment. We would like to emphasize the non-trivial difficulty and computational cost of significantly changing an existing pipeline and tweaking it to obtain the best performance, and because of this we leave comprehensive investigation of applying our techniques to non-SoTA frameworks as future work. With that said, we strongly agree with you that the proposed regularization has great potential and applicability to other learning frameworks. Therefore, following your suggestion, we plan to include additional experiments where we apply the coding rate regularizer to some (simple and lightweight) frameworks such as SimSiam [4] and assess its effectiveness.
> > >
> > > We will put all these results and any more results we obtain (i.e., about Barlow Twins loss and SimSiam) in the final version of the paper. Furthermore, we will publish our implementation upon the acceptance of our work to make it accessible for any researcher to reproduce our results and apply our method to their own problems or datasets. Again, we thank you for your constructive feedback and we hope our response resolves your remaining concerns. Please let us know if you have any further questions or comments.
> > >
> > > [1] Chen, Ting, et al. "A simple framework for contrastive learning of visual representations." International conference on machine learning, 2020.
> > >
> > > [2] Wang, Tongzhou, and Phillip Isola. "Understanding contrastive representation learning through alignment and uniformity on the hypersphere." International conference on machine learning, 2020.
> > >
> > > [3] Zbontar, Jure, et al. "Barlow twins: Self-supervised learning via redundancy reduction." International conference on machine learning, 2021.
> > >
> > > [4] Chen, Xinlei, and Kaiming He. "Exploring simple siamese representation learning." Proceedings of the IEEE/CVF conference on computer vision and pattern recognition. 2021.

---

### Official Review · Reviewer_5gcy · 2025-03-09

**Overall Recommendation:** 5

**Summary:**

The paper reveals that many comonents in the pipelines and models of DINO and DINOv2 are used to avoid collapse of the learned representations, and thus attempts to simplify the training pipelines, architectures, and hyperparameters of DINO and DINOv2 by introducing a $\ell^2$ distance based on the loss function plus a total coding rate based regularization to replace the cross entropy based loss function.  Such a simplification leads to more simple, efficient, and robust training pipelines and models---SimDINO and SimDINOv2, without need to use several manually designed components in models and tricks in training pipelines.  Extensive experiments demonstrate that SimDINO and SimDINOv2 outperform the original DINO and DINOv2 on the downstream tasks, including image classification, object detection and segmentation, semantic segmentation, and video object segmentation.  Moreover, the experimental results also demonstrate the stability and robustness against hyper-parameters.

## update after rebuttal:
The reviewer read the responses from the authors and the other review comments, and believes that this submission is an excellent work and should be accepted.

**Claims And Evidence:**

The claims made in the paper is supported by extensive experiments on the downstream tasks, including image classification, object detection and segmentation, semantic segmentation, and video object segmentation.  Moreover, there is also partial theoretical analysis in the appendix to  reaveal the possible reasons of the training instability in DINO.

**Essential References Not Discussed:**

Essential references are discussed.

**Experimental Designs Or Analyses:**

Yes. The experimental designs are fair and sound.

**Methods And Evaluation Criteria:**

Yes. Simplifying DINO and DINOv2 does make a lot of sense for wide applications of unsupervised representation learning.

**Other Comments Or Suggestions:**

At L166 in right column: "Note that $d_{\ell^2}(z_c^{cls}, z_g^{cls}) = - (z_c^{cls})^\top z_g^{cls}$ since ....". This is not correct----a constant 2 is missing.

**Other Strengths And Weaknesses:**

Strengths:
+ Extensive experiments are provided to demonstrate the SimDINO and SimDINOv2 outperform the original DINO and DINOv2 on image classification, object detection and segmentation, semantic segmentation, and video object segmentation.
+ Experimental results also confirm the stability and robustness of  SimDINO and SimDINOv2 against hyper-parameters.

Weaknesses:
- There is no theoretical justification on the mechanism why adding the total coding rate maximization term leads to avoid the feature collapse. For example, what is the landscape of the optimal solutions when using the simplified loss function based on $\ell_2$ distance and the rate reduction regularization in SimDINO?

**Questions For Authors:**

Regarding to the mechanism of adding a total coding rate maximization term to avoid the collapse, is there possible to have some theoretical results? For example, what is the landscape of the optimal solutions when using the simplified loss function based on $\ell_2$ distance and the rate reduction regularization in SimDINO?

**Relation To Broader Scientific Literature:**

The papers simplified two very popular methods DINO and DINOv2 in unsupervised learning of visual representation and thus proposed two promising models: SimDINO and SimDINOv2. Since that DINO and DINOv2 are widely applied in visual representation learning and have broad connections to deep learning literature, the reviewer believe that the simplified models again have signifiant potential to advance the practice of visual representation learning.

**Theoretical Claims:**

Yes, the proofs in the appendix are correct.

---

> ### Author Rebuttal · Authors · 2025-04-01
>
> Dear Reviewer 5gcy,
>
> We deeply appreciate your insightful review and am pleased to see that you find our work backed by solid empirical studies, with potential to improve practice of visual representation learning. Here we attempt to address the concerns you posed.
>
> > Theoretical justification for coding rate regularization
>
> Thanks for your suggestion. It actually can be shown that when coding rate is maximized, it will induce the feature $Z$ to be full-rank. This can be proved by a spectral analysis as $\log\det(I+ Z^\top Z) = \sum_{i=1}^{\min(n,d)}\log(1+\sigma_i^2)$, where $\sigma_i$ is the $i$-th singular value of $Z$. Maximizing coding rate then leads to uniform non-zero singular values (as columns of $Z$ are $\ell^2$ normalized), making $Z$ full-rank. This means the learned features are spread on the unit sphere, avoiding a collapsed solution. On the other hand, the $\ell^2$ distance encourages the student features to be close to the teacher features on the same image, ensuring that we do not arbitrarily project different crops of the same image to large distances in the feature space. This is the extent of our added theoretical analysis for now; note that a full optimization theory-esque landscape analysis is very challenging and/or unreflective of practice due to the self-distillation (teacher/student) aspect, and we leave such analysis for future work. We hope these clarifications alleviate your concerns and we will put the relevant theoretical analysis in the final version of the paper.
>
> > The connection between MSE and dot product similarity
>
> Thank you for pointing out this minor algebraic error. Note that in in eq.(7) we have included a factor of $\frac{1}{2}$, so the correct additive constant is +1. We will amend this in the final version of the paper.
>
> We greatly appreciate your time in evaluating our work and your suggestions will certainly help improve the quality of our work. Please let us know if you have more questions or concerns and we are happy to address them.

---

> > ### Comment · Reviewer_5gcy · 2025-04-06
> >
> > The reviewer appreciate the author's effort to carefully address the concerns and agrees that the self-distillation involving student  and teacher networks makes the theoretical justification quite challenging.
> >
> > Due to the supportive experiments and the elegant way to address the feature collapse issue of DINO and DINOv2, the reviewer believes that this is an excellent work and should be accepted.

---

> > > ### Author Response · Authors · 2025-04-08
> > >
> > > We sincerely appreciate your positive evaluation of our work and your valuable suggestions in improving our paper. Please let us know if you have any further comments or questions.

---

### Official Review · Reviewer_nrJd · 2025-03-15

**Overall Recommendation:** 4

**Summary:**

The work aims to simplify the architectures of DINO and DINOv2 family of models by adding an explicit coding rate term in their loss. The simplified architectures (referred as SimDINO and SimDINOv2) are more robust to different design choices and offer a Pareto improvement over the DINO model families. The simplification, if effective, could be useful for the larger field of AI and encourage more computationally efficient approaches.

**Claims And Evidence:**

The contribution of this work can be briefly summarized as: methods to reduce complexity of DINO and DINOv2 families by reducing complexities in training pipelines and adding a coding error rate regularizer to enable simple, robust and efficient training pipelines which result in an improvement performance in vision SSL.

**Essential References Not Discussed:**

I am not aware of any missing essential references.

**Experimental Designs Or Analyses:**

- Experiments compare the DINO approaches with their SimDINO alternatives on
    - Classification: Improvements in the performance have been highlighted in Table 1.
    - Object Detection and Segmentation: Tables 2 and 3 have highlighted the performance improvements for SimDINO.
- Segmentation maps for DINO and SimDINO in the supplementary material are comparable.

**Methods And Evaluation Criteria:**

- In Figure 1, it is stated that the simplification lies in removal of a linear layer and a softmax layer in DINO, and in the removal of the same layers along with one component of loss in DINOv2.
- The DINO pipeline is recapped (this is good for the readers) to highlight z^{cls}_c and z^{cls}_g obtained from local (student network) and global views (teacher network). It is argued that these features can be compared directly. However, the post-processing steps that are applied to prevent collapse result in a Loss highlighted in Equation 4. The complexities in updates of student and teacher parameters are further highlighted. It is explained how the collapse is avoided during EMA update while mentioned that the exact mechanism is not clear.
- The two main questions addressed are 'how to use z^{cls}_c and z^{cls}_g directly' and 'how to enforce non-collapse by efficient use of negative samples?'. The two questions, if addressed, could lead to removal of linear and softmax layers along with EMA update structure.
Equation (9) is proposed as a solution (for DINO) here which applies squared Euclidean distance and a rate distortion term with a hyperparameter γ and similarly Equation (17) has been proposed for DINOv2. Guidance on the hyperparameter γ has also been provided.
- The overall problem is well posed and the problem is formulated here.
- For evaluation, The ImageNet-1K classification dataset has been used for all methods for ViT backbones pretrained on DINO and SimDINO variants and COCOval2017, ADE20K and DAVIS-2017 for downstream detection and segmentation tasks.

**Other Comments Or Suggestions:**

- The last part 'vision SSL' (page 2, line 88) could perhaps be replaced by DINO and DINOv2 families or more details can be provided on how it would apply to the larger vision SSL field.
- The coding rate regularisation has been significantly used in the literature as mentioned in the Subection 'Coding rate, and related regularizers'. Since, the coding rate term (in Equation 9, 17) is central to the contribution, it would be good to compare variants of coding rate terms in literature and how they differ in the implementation, evaluation, etc.
- The improvements have been proposed over the existing DINO and DINOv2 pipelines. It would be good to summarize all the framework, training style variants of DINO and DINOv2 families which differ from the standard frameworks chosen for comparison. The contribution can be split into efficiency and improvement. Both of these are valuable. However, discussion and extension of experiments on design choices could strengthen the work. The literature and experiments could potentially be improved to include other common strategies that avoid representation collapse and how the proposed strategy is not only more efficient at avoiding the collapse but also more effective.
- Discussion on tradeoffs and efficiency could be provided when using alternate strategies for representation collapse.

**Other Strengths And Weaknesses:**

The manuscript is well-written and maintains high standards for readability. The problem and framework are well posed. Experiments have been conducted to highlight the value of proposed solutions. The design designs could potentially be further elaborated on by extending the experiments to include more variants of regularizers (including choices for coding rate terms), methods for improving efficiency, etc. This would help in establishing the value of proposed solution over the existing choices which could motivate the users for adoption.

**Questions For Authors:**

- The use of 'pareto improvement' term in the abstract is a slightly confusing to me. It is generally used if the improvement is coming at a cost of something (like a decrease) else (based on the Pareto definitions). In this work, I see that efficiency and performance are argued to improve together.
- I have shared my comments in other sections.

**Relation To Broader Scientific Literature:**

DINO families have been introduced with the background of contrastive SSLs and the issue of representation collapse is discussed in the context of contrastive SSL.  The use of negative samples (directly) in DINOv2 and (indirectly) in DINO models is highlighted along with the complexity and instability of the models along with the requirement of 'many tweaks' and careful hyperparameter selection for convergence.  The references to the corresponding sections  and/or papers should be cited while mentioning the issues with complexity of these models and the claim that all these are required for convergence. For instance, the appendix F.1 highlights some ablation studies on the stability of DINO training could be referred, etc. This work removes the 'many tweaks and hyperparameters' (figures, discussion could be referred accordingly) from DINO families by adding a coding error rate regularizer in the optimization.

**Theoretical Claims:**

Theoretical details have been listed for hyperparameter tuning in the Appendix C provided specifically to justify the choice of γ. I have not carefully verified the proof but the steps are consistent and I have not noticed any discrepancies in a brief look.

---

> ### Author Rebuttal · Authors · 2025-04-01
>
> Dear Reviewer nrJd,
>
> Thank you for your insightful comments and suggestions.  We are encouraged by your compliments on the simplification of the proposed method as well as the empirical evidence and presentation quality of our work. Here we attempt to address the concerns raised in your review.
>
> > Including more variants of regularizers and coding rate terms
>
> We appreciate your suggestion. First, we use coding rate as the anti-collapse loss here due to its simplicity and theoretical grounding (also refer to our response to Reviewer 5gcy for further theoretical justification). In terms of its computation, we would like to clarify that there do not exist many variants (i.e. all the cited works compute the exact same quantity). Other than coding rate, we agree it is possible to prevent collapse with other choices of regularizers and is an interesting direction to explore. We have experimented with directly using contrastive loss, but the results lag behind our choice. We will include these ablations in the camera-ready version of the paper.
>
> > Other variants of DINO and DINOv2 pipelines
>
> To the best of our knowledge, there aren’t any major variants of DINO and DINOv2 pipelines. Generally people either directly use the pretrained checkpoints or rely on the official implementations. We believe this can also be largely attributed to the complexity and fragility of the DINO pipeline, as we have demonstrated in Table 5 in the appendix. To compare our methods with the official pipeline, Figure 1 directly compares and summarizes the relevant changes.
>
> > Some issues on presentation
>
> Thank you for your comments. Regarding the use of "pareto improvement", we respectfully assert that a "Pareto improvement" is when the whole Pareto curve is moved forward (i.e., when improvements are gained without any cost). You may have been thinking of moving along the Pareto frontier, which involves tradeoffs. For "vision SSL" (page 2, line 88), we want to suggest that simplification can be very valuable to the larger field of vision SSL, and our work pushes this broader envelope. Rather than adding more tricks and complexities, we show that identifying the necessary components (i.e. feature alignment while avoiding collapse) naturally leads to a greatly simplified design and improved performance. We believe such practices are useful for the larger field of vision SSL. If this causes confusion, we are happy to revise it. We hope these clarifications can resolve your concerns.
>
> We again thank you for your time and effort in evaluating our work, which will no doubt improve its quality. We welcome your feedback and will be happy to engage in further discussion.

---

> > ### Comment · Reviewer_nrJd · 2025-04-06
> >
> > I thank the authors for responding to my comments and providing clarifications accordingly. My concerns regarding the use of "pareto improvement" and DINO pipelines are addressed. However, it would still be good to provide brief explanations on the alternate choice of regularizers for preventing collapse. The authors have mentioned the comparison of 'directly using contrastive loss' with their approach. It would be good to mention what variants of contrastive loss have been used and a brief summary of the comparison itself. In my opinion, that would make the contribution stronger. I am willing to change my initial score if brief summaries of the comparison results (and a summary of possible regularizers) to be included in the updated manuscript are provided.

---

> > > ### Author Response · Authors · 2025-04-08
> > >
> > > We are grateful for your review and valuable suggestions. We here provide you a summary of what we have done so far in verifying the effectiveness of our proposed coding rate regularizer. Recall that we regularize the student features on the global views in our experiment via the coding rate objective. As alternatives to our design, we have since tested on three different regularizers (with other settings fixed to be the same as SimDINOv2 on ViT-L/16) and we include their KNN accuracy results below:
> > >
> > > - the vanilla contrastive objective (i.e. explicitly repelling negative samples and attracting positive ones) as in SimCLR[1]. Compared to our result, this setting results in performance decrease of about 2 points (79.4 vs 81.1).
> > > - the uniform loss proposed in [2] that encourages the representation to be uniform on the unit sphere. This settings leads to performance decrease of about 4 points (77.2 vs 81.1).
> > > - Barlow Twins loss proposed in [3] that penalizes off-diagonal terms while promoting the on-diagonal terms on the covariance matrix of learned representations. This setting causes NaN in our experiments so far. It might be related to the sensitivity of the coefficient within the loss (as it needs to balance the off-diagonal and on-diagonal terms) and we are still investigating.
> > >
> > > We will put all these results and any more results we obtain (i.e., about Barlow Twins loss) in the final version of the paper.
> > >
> > > Overall, our experiments show that the coding rate is indeed a good choice in our setting, potentially due to the variance reduction reason proposed in the paper. We sincerely hope these results can help resolve your concerns.
> > >
> > > [1] Chen, Ting, et al. "A simple framework for contrastive learning of visual representations." International conference on machine learning, 2020.
> > >
> > > [2] Wang, Tongzhou, and Phillip Isola. "Understanding contrastive representation learning through alignment and uniformity on the hypersphere." International conference on machine learning, 2020.
> > >
> > > [3] Zbontar, Jure, et al. "Barlow twins: Self-supervised learning via redundancy reduction." International conference on machine learning, 2021.

---

### Official Review · Reviewer_S9wV · 2025-03-24

**Overall Recommendation:** 2

**Summary:**

The authors aim to improve the stability of self-supervised models DINO and DINOv2, while simplifying the training process. Collapse is a common problem in self-supervised learning (SSL), and it is usually avoided with careful hyperparameter tuning and architectural adaptations. In particular, to avoid collapse, DINO uses softmax temperature and centers the Teacher’s features, while DINOv2 adds KoLeo regularizer to make the span of features more uniform. The authors propose the following two main changes, (1) remove the heads, and replace cross-entropy loss with MSE on the $l_2$ normalized features of the encoder, (2) add rate distortion regularization to reduce the correlation between feature dimensions, which mitigates a constant output.

The authors experiment with ViT-S, B, and L models, which they pre-train on ImageNet-1K (IN-1K), and test on multiple downstream tasks, including classification, object detection, semantic segmentation, and video object segmentation. The proposed models, SimDINO and SimDINOv2, consistently outperform the DINO and DINOv2 baselines trained by the authors. They also show that hyperparameters from training ViT-B can be transferred to ViT-L, allowing for stable training. Finally, the authors provide ablations in the Appendix to study the effect of different hyperparameters and design choices, e.g., batch size.

## update after rebuttal
Because of the discrepancy in the description of the DINO loss, which is even involved in derivations, I can not recommend the paper for publication. However, the authors clarified/addressed a number of comments from my initial review, so, I increased my score from 1 to 2.

**Claims And Evidence:**

The authors make two main claims, (1) SimDINO and SimDINOv2 allow for simpler, more robust, and more computationally efficient training compared to DINO and DINOv2, (2) learned representations are of higher quality because of performance on downstream tasks.

- Performance: In the conducted experiments, the proposed models outperform the baselines across different downstream tasks, and different pre-training datasets. However, I see a number of issues with the experimental design:
    - Training duration: In Table 1, all models are trained for 100 epochs, while in the DINO paper [1] they report performance after 800 epochs, and they train for 100 epochs only for ablations. Having said that, the proposed method may be able to outperform the baselines if all models are trained longer, but this should be verified experimentally, especially since SSL methods benefit from extended training.
    - The provided experiments are adequate to show relative differences between DINO and SimDINO models, but additional baselines are required to be able to estimate the impact of the proposed method in the literature.
- Simplicity:
    - One of the experiments in support of the increased training stability is provided in Fig. 2, where SimDINO performance increases as training progresses, while DINO saturates and even experiences a decrease in accuracy. However, these results contradict the results reported in the original DINO paper, e.g., in Appendix Section D in [1], ViT-S improves its k-NN accuracy by a big margin as it is trained for 100, 300 and 800 epochs. This raises concerns about the training of the baselines, or about the adequacy of the training duration, since in Fig. 2 the models are trained only up to 100 epochs.
    - The authors claim that training is more efficient, however, training compute is not reported. The proposed pipeline is indeed simplified because the heads are removed, but a new loss term is introduced, which requires non-trivial computations. So, I think any gains in training efficiency should be quantified.

[1] Caron, Mathilde, et al. "Emerging properties in self-supervised vision transformers." *Proceedings of the IEEE/CVF international conference on computer vision*. 2021.

**Essential References Not Discussed:**

There are references that could be added, e.g., [4], but in general, I think the authors provide adequate references.

**Experimental Designs Or Analyses:**

- One of the main contributions of DINOv2 paper is performance at scale, both in terms of model and training data size. Based on that, the scale of the conducted experiments doesn’t allow for direct comparisons with the main claims of the DINOv2 paper.
- After pre-training, the authors keep the Teacher model and discard the Student (ln 188-189, col 2), however, in DINO paper the opposite holds true. Which model is kept for the reported baselines? Are there any noticeable differences in the performance of the Teacher and Student models?
- It would be interesting to have experiments with CNN backbones, similar to the DINO paper, to see the effect of the proposed regularizer, however, I don’t think this is a must-have, since DINO models are reported to excel with Transformer backbones.

**Methods And Evaluation Criteria:**

I think there are a number of issues with the description of the methods.

- ln 92, col 1: $R^{D \times *}$ represents the space of finite sequences, but in $U_{T=1}^{\infty}$, $T$ goes to infinity.
- What is the reason for defining a $d-1$ dim space instead of a $d$-dimensional one? For example, I could see how something like that would make sense for the sequence dimension $N$, since $N-1$ is the number pf patches, and there is an additional cls token for a total sequence length of $N$, but why for $d$? Also, in ln 133, col 1, the heads are defined as functions $R^d \rightarrow R^m$, but for their inputs $z$, in ln 166, col 2, we have $z \in S_{d-1}$.
- ln 99: $\Theta$ is not defined.
- ln 101: In the definition of $f_{\theta}^{\text{cls}}$, I guess it should be $S_{d-1}$ instead of $S^{d-1}$.
- ln 113-116, col 1: I think it would be good to make a distinction between a transformation that augments an image, and the output of such a transformation for a given image, which can be called a view. In ln 92-93, col 2, the term “view” is treated as the output of a transformation, i.e., “the feature corresponding to any view”, and in ln 113-116, col 1, as the transformation itself. The same in ln 121, col 1, where $X_c$ is called a “view”, while in the paragraph before, $v_c$ is defined as a “view””, and $X_c = v_c(X)$. SimCLR [2] has simple and clean notation for this issue.
- Eq. 2, 3: The temperature values should be different, because they are not assumed the same in DINO [1].
- ln 112, col 2: It is mentioned “the expectation is over $X$”, while there is no $X$ in Eq. 4.
- If I am not mistaken, the definition of DINO loss in Eq. 4, 5, 6 is incorrect. As can be seen in Algorithm 1 in [1], loss = H(t1, s2)/2 + H(t2, s1)/2 which is equivalent to $1/2 (CE(p_t(t_1), p_s(s_2)) + CE(p_t(t_2), p_s(s_1)))$, while based on Eq. 5 should hold $p_t(t_1) = p_s(s_1)$ and $p_s(s_2) = p_t(t_2)$, which is not true.
    - This affects derivations in Appendix B, where Eq. 5 is used.
    - Is this the loss used for DINO in the experiments?
- Fig 1: In (a), it seems that only the teacher weights are updated through EMA, but this is true for the head weights as well.
- Eq. 8 is not adequately explained. For example, in [3], which is cited by the authors, it is explained that “rate distortion $R(z, \epsilon)$ is the minimal number of binary bits needed to encode $z$ such that the expected decoding error is less than $\epsilon$”. This is the core contribution of the work, and the authors introduce $\epsilon$ as just a hyperparameter, without explaining its meaning.
- Footnote 3, starts at ln 215, col 2:
    - The authors refer to a similarity term in Eq. 9. I guess they refer to the 1st term, however it is an MSE term, not similarity term.
    - Their argument makes sense when the MSE is small, however, what if it is large?
- ln 245-246, col 1: $Z^{\text{patch}}$ and $P^{\text{patch}}$ are mentioned, e.g., “$i^{\text{th}}$ column of $Z^{\text{patch}}$”, before they are defined.

[2] Chen, Ting, et al. "A simple framework for contrastive learning of visual representations." *International conference on machine learning*. PmLR, 2020.

[3] Yu, Yaodong, et al. "Learning diverse and discriminative representations via the principle of maximal coding rate reduction." *Advances in neural information processing systems* 33 (2020): 9422-9434.

**Other Comments Or Suggestions:**

There are some minor typos, for example:

- ln 163, col 1: “receives a interpolated”, should be “an”.
- ln 112, col 1: “During the pipeline”, this phrase doesn’t seem right.
- ln 113-116, col 1: It is mentioned “we sample at random a view”, and then “selected randomly”, which seems unnecessary.

**Other Strengths And Weaknesses:**

No additional comments.

**Questions For Authors:**

No additional questions.

**Relation To Broader Scientific Literature:**

The proposed method builds directly on the DINO line of models, and offers increased stability and simplicity to the training process by introducing a rate distortion regularizer. As I mentioned earlier, this regularization term aims to reduce the correlation between feature dimensions, which is an idea already explored in SSL [4]. As a result, the proposed method is a combination of existing ideas in a modern setting, which I think has value, since SSL is an important research direction.

[4] Zbontar, Jure, et al. "Barlow twins: Self-supervised learning via redundancy reduction." *International conference on machine learning*. PMLR, 2021.

**Theoretical Claims:**

The authors provide theoretical results in Appendix Section B and C. I commented about Section B in the previous section. I went through the proof in Section C, but not in enough detail to be absolutely certain that is correct.

---

> ### Author Rebuttal · Authors · 2025-04-01
>
> Dear Reviewer S9wV,
>
> Thank you for the detailed review. Many comments, especially about presentation or typos, are helpful and will be incorporated in the camera-ready version; we do not address them here. Others require clarification or even stem from misunderstandings on your end, and we reply to these as follows.
>
> ## Claims and Evidence
>
> > Training duration
>
> Longer training usually leads to better performance, which is demonstrated in Appendix F.3. SimDINO still outperforms DINO with more training epochs. We are limited by compute and cannot adopt the exact same configurations as the original DINO(v2). With that said, we have since trained a ViT-B SimDINO model for 400 epochs with better performance in $k$-NN accuracy (76.6) than the official DINO model (76.1). This is still unfair comparison since the official setting uses full fp32 precision while ours only uses fp16 due to memory constraints. We will include more such results in our final version.
>
> > Additional baselines
>
> Please see our response to Reviewer ubYD (5-1).
>
> > Training stability
>
> It may seem contradictory but is actually not. For example, checkpoints at 100-th epoch in a 100-epoch run are very different from those in an 800-epoch run, due to difference in learning rate, teacher momentum, etc., and are incomparable. We also notice a similar performance dip in DINO trained for 200 epochs.
>
> > Computational efficiency
>
> The coding rate computation is efficient since covariance matrices are PSD. We will include quantitative analysis in the final version.
>
> ## Methods and Evaluation Criteria
>
> > Definition of $\mathbb{R}^{D \times *}$
>
> $\bigcup_{t = 1}^{\infty}\mathbb{R}^{d \times t} = \{x \colon \exists t\ \text{s.t.}\ x \in \mathbb{R}^{d \times t}\}$ is the set of all finite-size matrices with $d$ rows. This is a standard definition for the symbol $\bigcup_{t = 1}^{\infty}$.
>
> > Definition of $\mathbb{S}_{d - 1}$
>
> - The sphere in $\mathbb{R}^{d}$ is a $(d - 1)$-dimensional submanifold of $\mathbb{R}^{d}$. The definition including "$\mathbb{S}_{d - 1} \subseteq \mathbb{R}^{d}$" disambiguates this notation.
>
> > Definition of $\Theta$
>
> Here $\Theta$ is the weight space. We will make this more explicit.
>
> > Multiple definitions of view
>
> As is relatively common in SSL, we identify the function $v$ and the object $v(X)$ with each other for conceptual ease. We will make this more explicit.
>
> > Expectations over $X$
>
> The expectation is over $X$; the variables $z, p$, etc., are functions of $X$. We plan to streamline this notation to show dependencies.
>
> > Typo in the loss definition
>
> Thank you very much for pointing it out. We emphasize that **this is only a typographical error, and we use the official DINO(v2) losses in experiments**. We will fix this in the paper. However, it makes _very little_ difference in the main body and Appendix B, as the symmetry is not important for our argument.
>
> > Parameter $\epsilon$
>
> In this work, we only care that a smaller $\epsilon$ makes the regularizer larger. However, for completeness we will add a short description of the information-theoretic role of $\epsilon$.
>
> > MSE and similarity
>
> Since the features are normalized on the sphere, an MSE term _is_ a similarity term.
>
> > Covariance approximation
>
> The provided argument applies only when the MSE term is small. Since we choose $\gamma$ to balance the terms’ relative sizes, a well-trained SimDINO(v2) model _will_ have a small MSE term (induced by the feature alignment loss term), so the argument will apply.
>
> ## Supplementary Material
>
> > Section name choice
>
> We use this section label precisely because there is no self-distillation in this setting; one just tries to minimize the distance between two views' features, instead of taking one model as a privileged teacher from which the other network is distilled.
>
> ## Experimental Designs and Analyses
>
> > Performance at scale
>
> Our experiments (up to ViT-L and on ImageNet-1k) use very popular/standard settings in vision SSL. They are by-no-means considered to be limited scale. We are bound by our compute resources, and leave further scaling-up of our approach as future work.
>
> > Teacher/student weights
>
> Your assertion is false; both DINO and our work use the teacher weights for evaluation. This is shown in the original DINO paper (Fig.6), where the teacher model consistently outperforms the student.
>
> > CNN backbones
>
> As you point out, DINO(v2) perform strongly with ViT models, which are our focus as well. In particular, DINOv2 does not provide training on CNN backbones either.
>
> ## Overall
>
> Thanks again for your valuable feedback. We will fix many raised issues about prose and typos in the camera-ready version. We hope that our responses clarify the strength and correctness of our work. Given that we have attempted to address all of your main concerns and clarified several critical misconceptions of yours, we humbly request that you reconsider your recommendation for this work, as we believe it has good value to the community.

---

> > ### Comment · Reviewer_S9wV · 2025-04-05
> >
> > I would like to thank the authors for answering to all my points. I think some of them are still not addressed, so, I would like to offer the following comments:
> > - DINO loss symmetry in Eq.5
> >     - I start with this issue because it is one of my main concerns. As I mentioned in my review, Eq. 5 has 2 distributions, $p$ and $q$, while there should have been 4. The authors claim that this is a typographical error, however, the symmetry that involves these 2 distributions in Eq. 5 drives the derivations in Appendix B, so, how is this a typo? Also, why this is not important in Appendix B, since Eq. 21 and Eq. 24 don't apply to DINO? The authors state that this is not important, but they don't explain why.
> > - Training stability
> >     - I accept the authors' argument that some of the DINO, and especially DINOv2, experiments are very computationally expensive and should not be expected for new methods to always reach that scale. Having said that, the particular experiments in Fig. 2 have to do with training stability, and the original DINO was trained for 800 epochs, while most SSL methods train for much longer than 100 epochs. So, I am not claiming that the favorable behavior of SimDINO wouldn't extend to longer training times, but that there is no sufficient evidence for this due to the limited number of training epochs.
> > - Computational efficiency
> >     - I trust the authors that calculation of the proposed regularization term doesn't add a significant overhead, however, I would appreciate it if they could share the quantitative analysis that they plan to include in the updated manuscript.
> > - Definition of $S_{d-1}$
> >     - My question had to do about the reason d-dimensional latent variables belong to a d-1-dimensional sphere after normalization. It may be totally justified, I just said that the reason is not clear to me and I asked why this is.
> > - MSE and similarity
> >     - Based on the authors' response, I think there is a misunderstanding here, so, to clarify, my intention was to make a distinction between similarity and distance metrics, e.g., cosine similarity and Euclidean distance. When two vectors become more similar, similarity metrics increase, and distance metrics decrease. MSE falls in the latter category, thus, I think it is appropriate to call it a distance metric instead of a similarity metric.
> > - Teacher/student weights
> >     - I apologize for mentioning that the reported experiments in the DINO paper are with the Student model, this was my assumption, given that this is not explicitly mention in the paper. After more careful inspection of the official DINO github repo, I can confirm that the Teacher model is the one used in the reported DINO results.
> >
> > In summary, as I mentioned in my review, I think the authors work on an important research direction, i.e., stability of established SSL methods, however, due to the aforementioned issues I don't have the confidence to change my score.

---

> > > ### Author Response · Authors · 2025-04-08
> > >
> > > Thanks for your detailed response. We will try our best to resolve the remaining confusion.
> > >
> > >
> > > > DINO loss symmetry (or lack thereof)
> > >
> > >
> > > The original symmetrized objective was an inaccuracy in the presentation that occurred several times in the text, and we (again) thank you very much for pointing it out. We call it a "typographical" error to emphasize the fact that **it does not affect the experiments at all**.
> > >
> > >
> > > We are not sure what you mean by 4 vs 2 distributions, and you may be thinking of some other SSL method. In DINO there is the general (local or global) view taken by the student and the global view taken by the teacher, and that is it --- our notation gets this right, and in particular agrees with the DINO paper [1], which writes the loss as [quoted]
> > > $$\sum_{x \in \\{x_{1}^{g}, x_{2}^{g}\\}}\sum_{\\substack{x^{\prime} \in V \\\\ x^{\prime} \neq x}}H(P_{t}(x), P_{s}(x^{\prime})),$$
> > > which clearly shows the variables that we use.
> > >
> > >
> > > In terms of whether or not the symmetrization matters for our analysis in Appendix B, note that the optimality conditions of the CE loss remain the same for the symmetrized and non-symmetrized versions (i.e., $p = q$ and both are one-hot vectors), and Appendix B is only concerned with the optimality conditions, so the error does not substantively impact our argument at all, and the fixes required are trivial.
> > >
> > >
> > > > Training stability
> > >
> > >
> > > As pointed out in our previous response, the dip in performance in DINO happened not only in the 100-epoch run, but in other settings with more training epochs as well. Overall, our analysis on training stability does not depend on the number of training epochs, as shown in the last paragraph "More on Stability and Robustness" in Section 3.2. Specifically, our evidence reveals the high sensitivity of hyperparameters for DINO, as well as the difficulty associated with adapting DINO for different architecture (e.g. collapse in ViT-L) and different datasets (Fig.4). These results have nothing to do with the number of training epochs and demonstrate the superior stability and robustness of our proposed method.
> > >
> > >
> > > > Computational efficiency
> > >
> > >
> > > We again thank you for your valuable suggestion and will include the efficiency analysis in our final version. Concretely, we will measure the FLOPs and wall-clock time of our loss function and compare them with the original DINO loss (together with the linear head).
> > >
> > >
> > > > Definition of $\mathbb{S}_{d - 1}$
> > >
> > >
> > > The $d - 1$ dimensional sphere $\mathbb{S}_{d - 1} \subseteq \mathbb{R}^{d}$ is the set of all unit-norm vectors  in $\mathbb{R}^{d}$.
> > >
> > > By definition, if you take a nonzero vector in $\mathbb{R}^{d}$ and normalize it, it ends up on $\mathbb{S}_{d - 1}$.
> > >
> > >
> > > > MSE and similarity
> > >
> > >
> > > Sure, we can disambiguate this, thanks for pointing it out. But we emphasize (as you rightly point out) there is no conceptual or mathematical issue.
> > >
> > >
> > > We hope that our replies have clarified any remaining confusions. In light of our responses, we would kindly request that you re-evaluate the work.
> > >
> > >
> > > [1] Caron, Mathilde, et al. "Emerging properties in self-supervised vision transformers." Proceedings of the IEEE/CVF international conference on computer vision. 2021.

---

### Decision · Program_Chairs · 2025-05-01

**Decision:**

Accept (poster)

**Comment:**

This paper presents SimDINO and SimDINOv2, simplified variants of the widely used self-supervised learning methods DINO and DINOv2, by introducing a coding rate regularization term to replace complex collapse-prevention mechanisms. The paper is well-motivated and makes a strong case for reducing architectural and training complexity while achieving better or comparable performance across a range of downstream tasks.

Strengths:
* clear motivation and contribution: the authors propose a principled simplification of DINO/DINOv2 via a theoretically grounded regularizer, yielding improved stability and performance.
* extensive empirical validation: evaluations across classification, detection, and segmentation tasks, including ablations and comparisons to alternate regularizers, support the claims.
* robust rebuttal: the authors proactively addressed reviewers’ concerns (e.g., experimental scale, baseline comparisons, and loss formulation discrepancies) and provided new results, including comparative tests against SimCLR, uniformity, and Barlow Twins losses.

Some concerns:
* reviewer S9wV raised concerns about loss notation, derivations, and experimental scale. While the authors provided clarifications, the reviewer remained unconvinced despite acknowledging the paper’s direction is valuable.
* reviewer ubYD appreciated the idea but found the evidence around the proposed regularization insufficient, though authors provided new experiments and comparisons in response.

Despite two reviewers maintaining weak rejects, their critiques center around presentation and experimental thoroughness rather than flaws in core methodology. The authors’ detailed rebuttals and new experiments significantly mitigate these concerns. This paper delivers a well-executed, impactful simplification of an important line of SSL work with promising theoretical and empirical support. The proposed simplification is practical, effective, and of clear interest to the self-supervised learning community.